# Kolmogorov–Arnold Transformer

**Xingyi Yang,  Xinchao Wang**[*]
National University of Singapore
`xyang@u.nus.edu, xinchao@nus.edu.sg`

## Abstract

Transformers are the cornerstone of modern deep learning. Traditionally, they use multi-layer perceptron (MLP) layers to mix channel information. In this paper, we introduce the Kolmogorov–Arnold Transformer (KAT), which replaces MLP layers with Kolmogorov-Arnold Network (KAN) layers to enhance model expressiveness. Integrating KANs into transformers, however, is no easy feat, especially when scaled up. Specifically, we identify three key challenges: (C1) *Base function*. The standard B-spline used in KANs is inefficient for parallel computing, slowing inference. (C2) *Parameter and Computation Inefficiency*. KAN requires a unique function for each input-output pair, leading to high computational cost. (C3) *Weight initialization*. The initialization of KANs is particularly challenging due to their learnable activation functions. To overcome the aforementioned challenges, we propose three key solutions: (S1) *Rational basis*. We replace B-spline functions with rational functions to improve compatibility with modern GPUs. By implementing this in CUDA, we achieve faster computations. (S2) *Group KAN*. We share activation weights across groups of neurons to reduce computation without sacrificing performance. (S3) *Variance-preserving initialization*. We initialize activation weights to maintain variance across layers. With these designs, KAT scales effectively and readily outperforms traditional MLP-based transformers. We demonstrate the advantages of KAT across various tasks, including image recognition, segmentation, detection, table classification, and graph classification. It consistently enhances performance over the standard transformer architectures of different model sizes.

## 1 Introduction

Transformers have become the *de facto* architecture in deep learning, widely adopted in computer vision (Dosovitskiy et al., 2021) and natural language processing (Vaswani et al., 2017). At their core, transformers rely on two main components: attention modules and multi-layer perceptrons (MLPs). Although significant research has explored replacing the attention mechanism (Liu et al., 2021; 2022; Tolstikhin et al., 2021), most variants still rely heavily on MLPs. Surprisingly, there have been few attempts (Shazeer, 2020) to improve MLPs themselves.

Opening up the box, MLPs consist of stacked linear layers with non-linear activations. Theoretically, they can approximate any function given enough neurons (Hornik et al., 1989). However, MLPs come with their own problems. They struggle to model complex functions, like fitting periodic patterns with ReLU-like activations. Training these networks often results in slow convergence for high-frequency components (Rahaman et al., 2019; Basri et al., 2020; Ronen et al., 2019). These limitations have driven researchers to explore more expressive alternatives to MLPs.

Recently, Kolmogorov-Arnold Networks (KANs) emerged as a powerful alternative. KANs are noted for their theoretical parameter efficiency, potentially requiring fewer parameters to model complex functions (Liu et al., 2024b; Yu et al., 2024; Bozorgasl & Chen, 2024a; Liu et al., 2024a). The key to such success is the learnable base functions for each input-output pair, often parameterized by B-spline curves (Unser et al., 1993; Gordon & Riesenfeld, 1974). Given its potential, integrating KANs into transformers (Vaswani et al., 2017) becomes an exciting topic.

---

[*]Corresponding Author

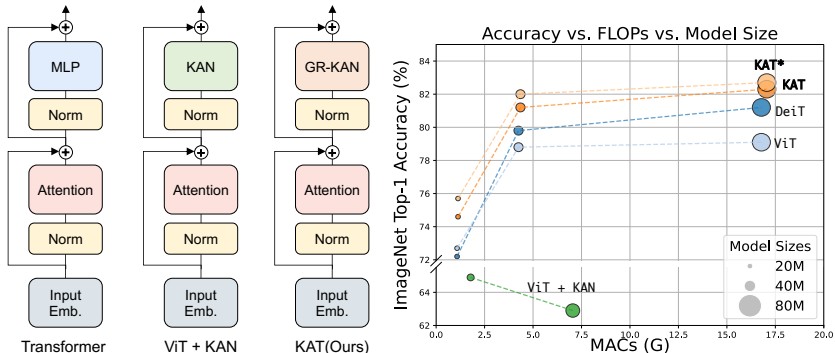

Figure 1: (Left) Architecture of standard transformer (e.g. ViT), ViT+KAN which replaces the MLP with a KAN, and our KAT model. In KAT, the MLP layers are replaced with GR-KAN layers. (Right) Performance on the ImageNet dataset. KAT* indicates that the model was initialized using a pre-trained ViT. Generally, KAT outperforms both the ViT and DeiT models. ViT+KAN performs poorly on ImageNet-level training.

Unfortunately, this ambition has been met with limited success. In particular, KANs have been reported to be "10× slower than MLPs, given the same number of parameters". Initial efforts in vision recognition tasks have been disappointing. Even on a small scale, these studies have consistently fallen short of matching, let alone surpassing, the performance of traditional architectures. This lack of improvement is often attributed to the limited computational resources and ongoing scalability problems (Cheon, 2024a; Bodner et al., 2024; Cheon, 2024a;b).

In a preliminary experiment, we replaced the MLP layers in the Vision Transformer (ViT) with KAN layers, creating the ViT+KAN model. As shown in Figure 1 (Right), this straightforward substitution led to significant challenges during ImageNet-scale training and resulted in poor performance. Scalability, therefore, remains a major obstacle for KAN-based models.

**Motivation and Challenges.** Through dedicated analysis, we have identified several key challenges that hinder the effectiveness of KANs in large-scale applications, ultimately limiting their scalability.

- **(C1)** *Base function.* Standard B-spline functions in KANs are incompatible with modern GPUs because they rely on recursive computations. This recursive nature significantly slows down performance, even in highly optimized implementations.
- **(C2)** *Parameter and Computation Inefficiency.* KANs require unique parameters and base functions for each input-output pair. It leads to exponential growth in parameters and computation overhead as the network scales.
- **(C3)** *Weight initialization.* KANs use the same weight initialization as MLPs. But this approach does not ensure convergence, leading to instability and degraded performance.

**Our Approach.** In this paper, we introduce *Kolmogorov–Arnold Transformer* (KAT), which successfully integrates KANs into transformers for large-scale training scenarios such as ImageNet. Beyond simple replacement, We have developed three key innovations (S1-S3) to address these challenges (C1-C3) respectively.

- **(S1)** *Rational activation.* We use rational functions as our base function with a full CUDA implementation, improving efficiency and compatibility with modern GPUs.
- **(S2)** *Group KAN.* We share function coefficients and base functions among groups of edges. This strategy reduces computational load significantly without sacrificing performance.
- **(S3)** *Variance-preserving initialization.* We carefully initialize the weights to keep the activation variance consistent across layers. This ensures stability during training and improves performance.

By combining solutions S1–S3, we present a new variant of KAN, called *Group-Rational KAN* (GR-KAN), to replace the MLP in transformer. GR-KAN is computationally efficient, easy to implement, and seamlessly integrates into transformers. Furthermore, our designs allow KAT to load pre-trained weights from ViT models and continue training to achieve even better results.

We empirically validate KAT across a range of vision tasks, including image recognition, object detection, and semantic segmentation. The results demonstrate that KAT outperforms traditional

MLP-based transformers, with similar computational requirements. As shown in Figure 1, KAT-B achieves 82.3% accuracy on ImageNet-1K, surpassing the ViT-B by 3.1%. When initialized with pre-trained weights from ViT, the performance further improves to 82.7%.

The contributions of our paper are threefold. First, we thoroughly analyze the challenges in scaling KAN-based models, particularly focusing on inefficiencies in base functions, parameterization, and weight initialization. Based on this analysis, we propose a set of solutions: rational activation functions tailored for GPU efficiency, Group KAN to reduce computational overhead, and variance-preserving initialization to ensure stable training. Second, leveraging these insights, we introduce the Kolmogorov–Arnold Transformer (KAT) and scale it to ImageNet-level training. Third, we validate our approach through extensive experiments, showing that KAT not only matches but surpasses the performance of ViT models, all under similar computational requirements.

## 2 PRELIMINARY

### 2.1 KOLMOGOROV-ARNOLD REPRESENTATION THEOREM

The Kolmogorov-Arnold representation theorem (Hecht-Nielsen, 1987) states that any multivariate continuous function $f$, defined in a bounded domain, can be expressed as a finite composition of continuous univariate functions and addition. Specifically, for a smooth function $f : [0,1]^n \to \mathbb{R}$, it can be represented as:

$$f(x_1, \ldots, x_n) = \sum_{q=1}^{2n+1} \Phi_q \left( \sum_{p=1}^{n} \phi_{q,p}(x_p) \right)$$

Here, each function $\phi_{q,p} : [0,1] \to \mathbb{R}$ and $\Phi_q : \mathbb{R} \to \mathbb{R}$ are continuous. This means that the (2d+1)(d+1) univariate functions $\Phi_q$ and $\phi_{q,p}$ are enough for an exact representation of a d-variate function.

This theorem can be written in matrix form as follows:

$$f(\mathbf{x}) = \Phi_{\text{out}} \circ \Phi_{\text{in}} \circ \mathbf{x} \tag{1}$$

where $\Phi_{\text{in}}$ and $\Phi_{\text{out}}$ are defined as:

$$\Phi_{\text{in}} = \begin{bmatrix} \phi_{1,1}(\cdot) & \cdots & \phi_{1,d}(\cdot) \\ \vdots & \ddots & \vdots \\ \phi_{2d+1,1}(\cdot) & \cdots & \phi_{2d+1,d}(\cdot) \end{bmatrix}, \quad \Phi_{\text{out}} = [\Phi_1(\cdot) \quad \cdots \quad \Phi_{2d+1}(\cdot)] \tag{2}$$

This decomposition illustrates how $f$ can be built from simpler functions, showcasing an essential property of multivariate continuous functions.

### 2.2 KOLMOGOROV–ARNOLD NETWORKS

Inspired by the Kolmogorov-Arnold representation theorem, (Liu et al., 2024b) define a generalized Kolmogorov-Arnold layer to learn a univariate non-linear function for each edge. By summing these univariate functions, the layer can approximate any multivariate function. Formally, a Kolmogorov-Arnold layer with $d_{\text{in}}$-dimensional inputs and $d_{\text{out}}$-dimensional outputs is illustrated as

$$f(\mathbf{x}) = \Phi \circ \mathbf{x} = \left[ \sum_{i=1}^{d_{in}} \phi_{i,1}(x_i) \quad \cdots \quad \sum_{i=1}^{d_{in}} \phi_{i,d_{out}}(x_i) \right], \text{where} \quad \Phi = \begin{bmatrix} \phi_{1,1}(\cdot) & \cdots & \phi_{1,d_{\text{in}}}(\cdot) \\ \vdots & \ddots & \vdots \\ \phi_{d_{\text{out}},1}(\cdot) & \cdots & \phi_{d_{\text{out}},d_{\text{in}}}(\cdot) \end{bmatrix} \tag{3}$$

Note that Eq 3 can be seen as a generalized form of Eq 1, such that $\Phi = \Phi_{\text{in}} \circ \Phi_{\text{out}}$. A general KAN network is a stacking of $L$ layers: given an input vector $\mathbf{x}_0 \in \mathbb{R}^{d_0}$, the output of KAN is $KAN(\mathbf{x}_0) = \Phi_{L-1} \circ \Phi_{L-2} \cdots \circ \Phi_0 \circ \mathbf{x}_0$.

In practice, (Liu et al., 2024b) parameterizes $\Phi$ use a linear combination of SiLU activation (Elfwing et al., 2018) and a B-spline function

$$\phi(x) = w_b \texttt{silu}(x) + w_s \texttt{spline}(x), \text{where} \quad \texttt{silu}(x) = \frac{x}{1+e^{-x}}, \quad \texttt{spline}(x) = \sum_i c_i B_i(x) \tag{4}$$

## 3  WHY ORIGINAL KAN FAILS TO SCALE?

This section examines the scalability of KAN. It is hindered by three factors: the choice of base function, redundant parameters and computations, and initialization problems. These design choices make the vanilla KAN resource-intensive and difficult to apply to large-scale models.

**B-spline is not GPU Friendly.**  The use of B-spline functions in KAN layers introduces challenges when implemented on GPUs. **First**, B-splines are not standard functions in CUDA, so using them via PyTorch or NumPy lacks optimized support, resulting in slower performance. **Second**, the localized computations of B-splines complicate parallel processing: each control point affects only a small area, leading to sparse or recursive operations that GPUs handle less efficiently. While efficient implementations exist for cubic B-splines (Ruijters & Thévenaz, 2012; Ruijters et al., 2008; Sigg & Hadwiger, 2005), scaling these methods to higher orders is not straightforward.

**Parameter and Computation Inefficiency.**  KAN differs from standard neural networks by using a learnable base function for each input-output channel pair. This design increases the number of parameters and computational demands, especially as the network's width and depth grow.

In a typical KAN layer with $d_{in}$ input and $d_{out}$ output channels, a B-spline function of order $K$ over $G$ intervals is assigned to each input-output pair. This results in $(d_{in} \times d_{out}) \times (G + K + 3) + d_{out}$ learnable parameters. In contrast, a standard MLP requires only $(d_{in} \times d_{out}) + d_{out}$ parameters.

For computation (Yu et al., 2024), the FLOPs required to process one sample using the De Boor-Cox iterative formulation (Boor, 1971) are $\Big\{ \text{FLOPs of non-linear function} \times d_{in} + (d_{in} \times d_{out}) \times [9K \times (G + 1.5K) + 2G - 2.5K + 3] \Big\}$. In contrast, the FLOPs for an equivalent MLP layer is merely $\Big\{ \text{FLOPs of non-linear function} \times d_{out} + 2 \times (d_{in} \times d_{out}) \Big\}$.

Overall, KAN's parameter size and computational effort are $O(G + K)$ and $O(GK)$ times greater than those of a conventional MLP, respectively. This makes scaling up KAN challenging.

**Weights are not Properly Initialized.** Proper weight initialization is essential for training deep neural networks. A fundamental principle is *variance-preserving*, meaning that the variance of the signal should remain constant as it propagates through multiple layers, whether forward or backward (LeCun et al., 2002; Glorot & Bengio, 2010; He et al., 2015). This variance-preserving principle stabilizes activations and gradients.

However, the KAN paper's initialization strategy violates this principle. Assuming an input $x \sim \mathcal{N}(0, \sigma_x^2)$ and using a zero-order spline, the output variance becomes $Var[\phi(x)] \approx 0.01 + 1.064\sigma_x^2$[1], which differs from $Var[x]$. Higher-order splines may exacerbate this variance instability. Therefore, the default initialization contradicts the essential variance-preserving principle.

## 4  KOLMOGOROV–ARNOLD TRANSFORMER

As discussed earlier, the standard KAN faces three major challenges that limit its use in large, deep neural networks. In this section, we refine its design to better suit modern transformers, allowing us to replace MLP layers with KANs.

### 4.1  OVERALL ARCHITECTURE

Just as its name imply, Kolmogorov–Arnold Transformer (KAT) replaces the MLPs in vision transformer (Dosovitskiy et al., 2021) with KAN layers.

Specifically, for a 2D image $\mathbf{x} \in \mathbb{R}^{H \times W \times C}$, we first flatten it into a 1D sequence, apply patch embedding and positional encoding, and then pass it through a series of KAT layers. At layer $\ell$, the following operations are performed:

$$\mathbf{x}_0^{(\ell)} = \text{MSA}(\text{LN}(x_{\ell-1})) + \mathbf{x}_{\ell-1}, \quad \ell = 1, \ldots, L \tag{5}$$

---

[1]full derivation in Appendix D.

$$\mathbf{x}_\ell = \text{MLP}(\text{LN}(\mathbf{x}_0^{(\ell)})) + \mathbf{x}_0^{(\ell)}, \quad [\textbf{Transformer}] \quad (6) \qquad \mathbf{x}_\ell = \text{KAN}(\text{LN}(\mathbf{x}_0^{(\ell)})) + \mathbf{x}_0^{(\ell)}, \quad [\textbf{KAT}] \quad (7)$$

where $\mathbf{x}_\ell$ stands for the output feature sequence at the $\ell$ layer. MSA and LN stand for the multi-head self-attention and layer norm. As illustrated, we replace all two-layer MLPs with two-layer KANs while keeping the attention layers unchanged. Although similar efforts have been made in specific domains (Chen et al., 2024b;a), a simple replacement is not enough to achieve scalability in large models.

Most importantly, here, we introduce a special kind *Group-Rational KAN*. We use rational functions as the base function for KAN (Section 4.2) and share parameters between a group of edges (Section 4.3). We also specify the weight initialization scheme to ensure stable training (Section 4.4). Together, these enhancements make KAT more scalable and improve performance.

## 4.2 RATIONAL BASE FUNCTIONS

In our method, we use the rational function (Boullé et al., 2020; Telgarsky, 2017; Leung & Haykin, 1993; Aghaei, 2024) as the base function for the KAN layer, instead of the B-spline.

Specifically, we parameterize the function $\phi(x)$ on each edge as rational over polynomials $P(x), Q(x)$ of order $m, n$.

$$\phi(x) = wF(x) = w\frac{P(x)}{Q(x)} = w\frac{a_0 + a_1 x + \cdots + a_m x^m}{b_0 + b_1 x + \cdots + b_n x^n} \qquad (8)$$

$a_n$ and $b_m$ are coefficient of the rational function and $w$ is the scaling factor. This function is said to have degree $m/n$. We hope to learn those $a_n, b_m$ and $w$ through end-to-end backpropagation.

To avoid instability caused by poles, where $Q(x) \to 0$ and $\phi(x) \to \pm\infty$, we employ a Safe Padé Activation Unit (PAU) (Molina et al., 2020) as our basis, which is a modified form of the standard rational function

$$F(x) = \frac{a_0 + a_1 x + \cdots + a_m x^m}{1 + |b_1 x + \cdots + b_n x^n|} \qquad (9)$$

**Implement Rational Function on GPU.** With the rational function, a core contribution in this paper is to implement it efficiently on the GPU. Rather than using `pytorch` with automatic differentiation, we implement it fully with CUDA (Nickolls et al., 2008).

- Similar to (Molina et al., 2020), we compute the explicit gradients of $\frac{\delta F}{\delta a_m}, \frac{\delta F}{\delta b_n}$ and $\frac{\delta F}{\delta x}$. The full expression is shown in Appendix E.
- To optimize the evaluation of polynomials, we employ Horner's method (Horner, 1815), which reformulates a polynomial in a nested form to reduce the computation:

$$a_0 + a_1 x + \cdots + a_m x^m = a_0 + x(a_1 + x(a_2 + x(\ldots))) \qquad (10)$$

  This allows the evaluation of a polynomial of degree n with only $n$ multiplications and $n$ additions. By default, we use $m = 5$ and $n = 4$.

Through this efficient CUDA implementation, we largely reduce the computation for each evaluation of the base function. As shown in Table 1, with a scalar input, the rational function with the Horner method is much cheaper than the B-spline used in the KAN paper.

## 4.3 GROUP KAN

Instead of learning a unique base function for each input-output pair, we can share their parameters within a group of edges. It reduces the number of parameters and computation. This kind of parameter sharing (LeCun et al., 1995; 1989) and group-wise computation (Vaswani et al., 2017; Wu & He, 2018) have been key techniques in neural network design.

Specifically, we divide the input channels $d_{in}$ into $g$ groups, sharing parameters among $d_{in}/g$ input channels within each group. Figure 2 illustrates the distinctions between the original KAN, our Group KAN, and a standard MLP. Unlike MLPs, which employ non-learnable activations, KAN assigns a unique function to each input-output pair. Group KAN reduces the number of parameters by sharing these functions among a group of edges.

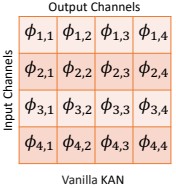

Figure 2: Comparing our Group KAN with vanilla KAN and MLPs. While KAN has unique function on each input-output pair, Group KAN shares these functions with groups of edges.

| Name | FLOPs |
|------|-------|
| B-Spline (G=3, K=3) | 204 |
| Rational (m=5, n=4) | 28 |
| Rational (m=5, n=4) w Horner | 21 |

Table 1: One sample FLOPs comparison of different non-linear functions at each edge. Horner's method with the Rational function reduces FLOPs by approximately $9.3\times$ compared to the B-Spline.

| Name | No. Params | FLOPs |
|------|-----------|-------|
| MLP | $d_{in} \times d_{out} + d_{out}$ | Func FLOPs $\times d_{out} + 2 \times (d_{in} \times d_{out})$ |
| KAN | $d_{in} \times d_{out} \times (G + K + 3) + d_{out}$ | Func FLOPs $\times d_{in} + (d_{in} \times d_{out}) \times [9K \times (G + 1.5K) + 2G - 2.5K + 3]$ |
| GR-KAN (Ours) | $d_{in} \times d_{out} + d_{out} + (m + n \times g)$ | $(2m + 2n + 3) \times d_{in} + 2 \times (d_{in} \times d_{out})$ |

Table 2: Comparison of parameter number and computation among different models. *Func FLOPs* refers to the FLOPs of non-linear activation. In KAN, $K$ is the order and $G$ is the grid number. In GR-KAN, $m$ and $n$ are the polynomial orders, and $g$ is the number of groups. GR-KAN has a parameter size comparable to that of MLP, while the original KAN has $O(G + K)$ times more parameters and $O(GK)$ times more FLOPs.

**Group-Rational KAN.** We combine the rational function of Section 4.2 with group-wise parameters to implement our Group-Rational KAN (GR-KAN). In practice, we share the parameter for the rational function $F$ for each group; however, each edge retains a unique scalar $w$.

Suppose $i$ is the index of the input channel. With $g$ groups, each group contains $d_g = d_{in}/g$ channels, where $\lfloor i/d_g \rfloor$ is the group index. The operation of GR-KAN on input vector $\mathbf{x}$ can be expressed as

$$\text{GR-KAN}(\mathbf{x}) = \Phi \circ \mathbf{x} = \left[ \sum_{i=1}^{d_{in}} w_{i,1} F_{\lfloor i/d_g \rfloor}(x_i) \quad \cdots \quad \sum_{i=1}^{d_{in}} w_{i,d_{out}} F_{\lfloor i/d_g \rfloor}(x_i) \right] \quad (11)$$

With a simple rewrite, this can be expressed in matrix form as the product of a weight matrix $\mathbf{W} \in \mathbb{R}^{d_{in} \times d_{out}}$ and a input-wise rational function $\mathbf{F}$

$$\text{GR-KAN}(\mathbf{x}) = \mathbf{W}\mathbf{F}(\mathbf{x}) = \begin{bmatrix} w_{1,1} & \cdots & w_{1,d_{in}} \\ \vdots & \ddots & \vdots \\ w_{d_{out},1} & \cdots & w_{d_{out},d_{in}} \end{bmatrix} \times \begin{bmatrix} F_{\lfloor 1/d_g \rfloor}(x_1) & \cdots & F_{\lfloor d_{in}/d_g \rfloor}(x_{d_{in}}) \end{bmatrix}^{\top}$$

$$(12)$$

As such, we can implement this GR-KAN layer as a group-wise rational function $\mathbf{F}$ followed by a linear layer

$$\text{GR-KAN}(\mathbf{x}) = \texttt{linear}(\texttt{group\_rational}(\mathbf{x})) \quad (13)$$

In this form, sharing parameters across each input channel allows direct application of the rational function to the input vector, equivalently applying it across each grouped edge. In this way, GR-KAN functions as a specialized MLP, with 1) learnable non-linear functions, 2) activation preceding the linear layer, and 3) unique activation functions tailored for each group of edges.

In experiments, we notice that for rational function, we share the numerator coefficient $a_m$ among all groups and use different $b_n$ for each group. It gets better performance.

**Parameter and Computation Savings.** The original KAN requires $d_{in} \times d_{out}$ unique activation functions. Through our grouping strategy, only $g$ unique functions are needed, reducing the parameter count to a constant overhead compared to a standard MLP.

Except the saving on parameter number, this grouping also reduces computational demands. Each input channel computes the activation function $\phi$ once, shared across all corresponding output channels. In contrast, the original KAN requires that each output channel $j$ to independently compute $\phi_{i,j}$. This results in significant computational savings. The comparison of the number of parameters and computation is listed in Table 2.

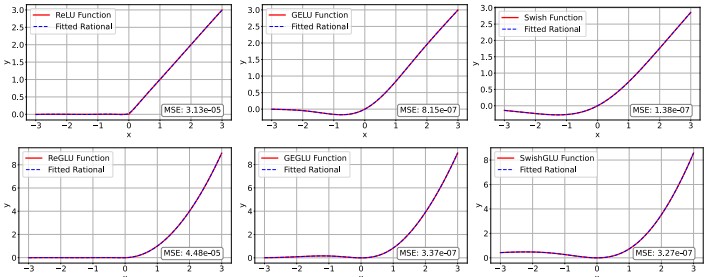

| Name | $\alpha = \frac{Var[x]}{\mathbb{E}[F(x)^2]}$ |
|---|---|
| Identity | 1 |
| ReLU | 2 |
| GELU | 2.3568 |
| Swish/SiLU | 2.8178 |
| GEGLU | 0.7112 |
| SwishGLU | 0.8434 |

Figure 3: Example of fitted functions with rational form. We also show the mean square error (MSE) for the fitting.

Table 3: Expected values of $F(x)^2$ for various functions.

## 4.4 VARIANCE-PRESERVING INITIALIZATION

In this section, we aim to initialize the values for $a_m, b_n$ and $w$ in Group-Rational KAN to ensure variance-preserving behavior across the network. At its core, we prevent the growth or reduction of activation magnitudes throughout the layers, thereby maintaining stability.

We revisit the analysis from (He et al., 2015) and adapt it to KANs. For a GR-KAN layer, the computation for each output $y_j$ is given by $y_j = \sum_{i=1}^{d_{in}} \phi(x_i) = \sum_{i=1}^{d_{in}} (w_{i,j} F(x_i)) + b_j$. We assume that all $x_i$ are *i.i.d* (Glorot & Bengio, 2010) and uniformly distributed. Here, $w_{i,j}$ follows a normal distribution $\mathcal{N}(0, \sigma_w^2)$ and $b_j$ is initialized to zero. The variance of $y_j$ can then be described as:

$$Var[y] = d_{in} Var[wF(x)] \tag{14}$$

$$Var[y] = d_{in} Var[w]\mathbb{E}[F(x)^2] \tag{15}$$

where $x$, $y$, and $w$ represent the random variables of each element in $x_i$, $y_j$, and $w_{i,j}$ respectively. When layers are stacked, we aim for the variance of the input-output activations to remain consistent, expressed as:

$$Var[x] = d_{in} Var[w]\mathbb{E}[F(x)^2] \tag{16}$$

Since $F(x)$ is the rational function containing coefficients $a_m$ and $b_n$, the initialization of $w$ and these coefficients are interdependent—the form of $F(x)$ influences the appropriate initialization of $w$. The crucial step is to calculate $\frac{Var[x]}{\mathbb{E}[F(x)^2]}$ and adjust $w$ to maintain consistent activation scaling.

For our rational function defined in Equation 9, computing $\mathbb{E}[F(x)^2]$ involves evaluating:

$$\mathbb{E}[F(x)^2] = \int_{-\infty}^{+\infty} F^2(x)f(x)dx = \int_{-\infty}^{+\infty} (\frac{a_0 + a_1 x + \cdots + a_m x^m}{1 + |b_1 x + \cdots + b_n x^n|})^2 f(x)dx \tag{17}$$

where $f(x)$ is the density function of $x$. Unlike activation functions such as ReLU, for which $\mathbb{E}[F(x)^2] = \frac{1}{2}Var[x]$, computing $\mathbb{E}[F(x)^2]$ for the rational function is challenging due to the lack of a closed-form solution.

**Initialize $a, b$ first, then initialize $w$.** To make the process manageable, Instead of sampling $w$, $a$, and $b$ jointly, we proceed sequentially. Initially, we determine $a$ and $b$ such that $F$ fits established activations like ReLU, GELU, and Swish. Figure 3 illustrates the fitted functions.

Once $a$ and $b$ are set, we estimate the gain $\alpha = \frac{\mathbb{E}[F(x)^2]}{Var[x]}$ numerically, assuming $x \sim \mathcal{N}(0, 1)^2$. The calculated gains, $\alpha$, are documented in Table 3. We use the gain value to initialize $w$ from $\mathcal{N}(0, \frac{\alpha}{d_{in}})$.

**Initialize KAT from ViT.** In addition to random weight initialization, we can also transfer weights from a pre-trained ViT to our KAT model. This transfer is straightforward for most layers, as KAT can replicate the micro-architecture of ViT, except for the KAN layer.

---

[2]This assumption is justified as the inputs to the KAN layer are normalized using layer normalization as in Equation 7. The LN layers are initialized to have zero bias and a scaling factor of one

For the GR-KAN, weight transfer is still feasible, as shown in Figure 4. Because the GR-KAN also includes a linear layer, we can directly load the weights of the linear layer from the MLP in the trained ViT.

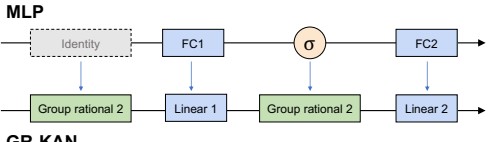

For rational layers in GR-KAN, the first one is initialized to behave like an identity function,

Figure 4: One-to-one weight mapping between trained MLP in ViT and GR-KAN in KAT.

while the second layer is set to approximate the non-linear function used in the original MLP. This approach allows all the weights of the GR-KAN layer to be cloned from a ViT model, ensuring compatibility and efficient initialization.

# 5 EXPERIMENTS

## 5.1 EXPERIMENTAL SETUP

We modify the original ViT (Dosovitskiy et al., 2021) architecture by substituting its MLP layers with GR-KAN layers. By default, these KAN layers employ a rational function with parameters $m = 5$ and $n = 4$, and are organized into groups of 8 ($g = 8$). Each transformer block contains 2 KAN layers. The first GR-KAN layer's $a_m$ and $b_n$ are initialized to fit the identity function, while the second is initialized to mimic the Swish function (Ramachandran et al., 2017). The attention layers are initialized with Mimetic Initialization (Trockman & Kolter, 2023). The remainder of the architecture remains unchanged. We intentionally do not use hierarchical architectures (Yu et al., 2022) for simplicity. In the main paper, due to the page limit, we focus on vision tasks. But we also extend the architecture for tabular and graph data, as detailed in Appendix A.

## 5.2 IMAGE RECOGNITION

**Experiment Setup.** We do experiments on ImageNet-1K [59] image classification benchmark. ImageNet-1K is one of the most widely-used datasets in computer vision which contains about 1.3M images of 1K classes on training set, and 50K images on validation set.

We mainly follow the hyper-parameters of DeiT (Touvron et al., 2021). Specifically, models are trained for 300 epochs at $224^2$ resolution. The patch size is set to 16. Data augmentation and regularization techniques include RandAugment (Cubuk et al., 2020), Mixup (Zhang et al., 2018), CutMix (Yun et al., 2019), Random Erasing (Zhong et al., 2020), weight decay, Label Smoothing (Szegedy et al., 2016) and Stochastic Depth (Huang et al., 2016). We adopt AdamW (Loshchilov & Hutter, 2019) optimizer with batch size of 1024.

We compare with ViT (Dosovitskiy et al., 2021) and DeiT (Touvron et al., 2021), as we share the same architecture, except for the channel mixer. We also report the results of ViT + KAN (Chen et al., 2024b), that simply replacing MLP with standard KAN.

**Results.** Our experimental results demonstrate that the KAT models consistently outperform their counterparts on the IN-1k dataset, as shown in Table 5. Firstly, the integration of GR-KAN in transformer demonstrates superior performance over traditional MLP-based mixers. For instance, the KAT-S achieves an accuracy of 81.2%, outperforming the DeiT-S model by 2.4%. This improvement underscores

Figure 5: Comparative Analysis of Model Performance and Computational Efficiency on ImageNet-1K. We measure the FLOPs under $224^2$ using `fvcore` package. ∗ indicates that the model is initialized using a pre-trained ViT model, otherwise trained from scratch.

| Model | Channel Mixer | #Param. | FLOPs | IN-1k Top-1 |
|---|---|---|---|---|
| ViT-Ti/16 | MLP | 5.7M | 1.08G | 72.7 |
| DeiT-T | MLP | 5.7M | 1.08G | 72.2 |
| ViT-T + KAN | KAN | 12.8M | 1.78G | 64.9 |
| KAT-T | KAN | 5.7M | 1.13G | **74.6** |
| KAT-T∗ | KAN | 5.7M | 1.13G | **75.7** |
| ViT-S/16 | MLP | 22.1M | 4.25G | 78.8 |
| DeiT-S | MLP | 22.1M | 4.25G | 79.8 |
| ViT-S + KAN | KAN | 50.4M | 7.05G | 62.9 |
| KAT-S | KAN | 22.1M | 4.35G | **81.2** |
| KAT-S∗ | KAN | 22.1M | 4.35G | **82.0** |
| ViT-B/16 | MLP | 86.6M | 16.87G | 79.1 |
| DeiT-B | MLP | 86.6M | 16.87G | 81.8 |
| ViT-B + KAN | KAN | 199.8M | 28.04G | NAN |
| KAT-B | KAN | 86.6M | 17.06G | **82.3** |
| KAT-B∗ | KAN | 86.6M | 17.06G | **82.8** |

the potential of KAN mixers to enhance model efficacy when properly integrated.

Secondly, the vanilla KAN layer faces scalability issues. ViT-T/S + KAN only achieved an accuracy of around 63%, even with a much higher computational cost. ViT-B + KAN fails to converge, resulting in `NAN` error. We addressed these scaling challenges as detailed in Section 3, enabling our KAT models to scale successfully. These findings highlight the advantage of KAT models in balancing computational efficiency with improved performance.

## 5.3 OBJECT DETECTION AND INSTANCE SEGMENTATION

**Experimental Setup.** We evaluate our approach on the MS-COCO2017 (Lin et al., 2014) dataset, a standard benchmark for object detection and instance segmentation. In our setup, the KAT is employed as the backbone within a ViTDet-based (Li et al., 2022) Mask R-CNN (He et al., 2017) model, initialized with weights pre-trained on ImageNet. We followed the standard $3\times$ training schedule, which consists of 36 epochs. The training images were resized to $800 \times 1333$ pixels. The AdamW optimizer (Loshchilov & Hutter, 2019) was used with a learning rate of 0.0001 and a total batch size of 16. Our implementation was based on the PyTorch and MMDetection (Chen et al., 2019) libraries, and we use FP16 precision to reduce training costs. The experiments were carried out on 4 NVIDIA H100 GPUs.

**Results.** Table 4 compares the performance of different backbones. KAT consistently outperformed other models, particularly in object detection, where it achieved a 3.0 $AP^{box}$ gain on the S-sized model and a 1.4 $AP^{box}$ gain on the L-sized model compared to ViTDet. The improvements were most pronounced in smaller models, where computational cost increased by only 1 GFLOPs. This shows that KAT offers better accuracy with minimal overhead.

Table 4: Performance of Mask-RCNN with different backbones on $3\times$ schedule.

| Backbone | #Param. | FLOPs | $AP^{box}$ | $AP^{box}_{50}$ | $AP^{box}_{75}$ | $AP^{mask}$ | $AP^{mask}_{50}$ | $AP^{mask}_{75}$ |
|---|---|---|---|---|---|---|---|---|
| PVT-Small | 44.1M | - | 43.0 | 65.3 | 46.9 | 39.9 | 62.5 | 42.8 |
| Swin-T | 48M | 267G | 46.0 | 68.1 | 50.3 | 41.6 | 65.1 | **44.9** |
| ConvNeXt-T | 48M | 262G | 46.2 | 67.9 | 50.0 | **41.7** | 65.0 | **44.9** |
| ViT-S | 43.8M | 423G | 44.0 | 66.9 | 47.8 | 39.9 | 63.4 | 42.2 |
| ViTDet-S | 44.5M | 423G | 44.5 | 66.9 | 48.4 | 40.1 | 63.6 | 42.5 |
| KATDet-S | 44.5M | 424G | **47.5** | **69.0** | **51.2** | 41.5 | **65.7** | 44.0 |
| ViT-B | 113.6M | 767G | 45.8 | 68.2 | 50.1 | 41.3 | 65.1 | 44.4 |
| ViTDet-B | 113.6M | 767G | 46.3 | 68.6 | 50.5 | 41.6 | 65.3 | **44.5** |
| KATDet-B | 113.7M | 770G | **47.7** | **69.1** | **51.6** | **41.6** | **65.9** | 44.3 |

## 5.4 SEMANTIC SEGMENTATION

**Experiment Setup.** We evaluated our KAT model on the ADE20K dataset (Zhou et al., 2017). This dataset comprises 150 semantic categories with 20,000 images in the training set and 2,000 in the validation set. For our experiments, we utilized KAT as the backbone for the UperNet framework (Xiao et al., 2018), initializing it with ImageNet pre-trained weights. The training was conducted using the AdamW optimizer (Loshchilov & Hutter, 2019) with a learning rate of 0.0001 and a batch size of 16, across 160,000 iterations. Our implementation was carried out using the PyTorch and mmsegmentation libraries, and the experiments were performed on two NVIDIA H100 GPUs. For comparison, we evaluated UperNet with other backbones, including DeiT, Swin Transformer, and ConvNeXt.

**Results.** Table 5 summarizes the segmentation results. Overall, KAT demonstrates a competitive improvement over plain ViT-based architectures, achieving a 2.4% improvement over DeiT-S and a 0.2% improvement over DeiT-B. This performance boost comes with a slight increase in computational cost, reflected in the higher FLOPs. Similar to the detection results, KAT shows more significant gains in smaller models. However, it still falls short compared to models with hierarchical architectures, such as ConvNeXt, which benefit from more efficient structural design.

## 5.5 ABLATION STUDY AND ANALYSIS

**GR-KAN v.s. Activation Function.** As GR-KAN can be considered as a special kind of MLP with group rational function, we do an ablation study to consider different types of activation for

Table 5: Performance of Semantic segmentation with UperNet on ADE20K validation set. Images are cropped to $512 \times 512$ for training. The MACs are measured with input size of $512 \times 2048$.

| Backbone | #Param. | FLOPs | mIoU (%) |
|---|---|---|---|
| Swin-T | 60M | 945G | 45.8 |
| ConvNeXt-T | 60M | 939G | 46.7 |
| DeiT-S | 57M | 1217G | 43.5 |
| KAT-S | 57M | 1219G | 46.1 |
| Swin-B | 121M | 1188G | 49.5 |
| ConvNeXt-B | 122M | 1170G | 49.6 |
| DeiT-B | 142M | 2007G | 47.2 |
| KAT-B | 142M | 2011G | 47.4 |

Table 6: Ablation on activation function, with ViT-Ti/16 Variant.

| Name | Learnable? | IN-1k Top-1 |
|---|---|---|
| GELU (Default) | No | 72.7 |
| ReLU | No | 72.8 |
| SiLU | No | 69.8 |
| PReLU | Yes | 73.2 |
| PAU | Yes | 73.6 |
| KAT-T | Yes | **74.6** |

MLP and compare with our GR-KAN. Superficially, we replace the activation function in MLP in ViT-Ti/16 to different kinds, including GELU (Hendrycks & Gimpel, 2016), ReLU (Fukushima, 1969), SiLU (Elfwing et al., 2018), PReLU (He et al., 2015) and PAU (Molina et al., 2020), and comparing them with KAT.

Table 6 summarizes the top-1 accuracy on the ImageNet-1k dataset for each activation function with ViT-Ti/16. The results indicate that traditional activation functions like ReLU and GELU perform similarly. Learnable activations like PReLU and PAU show an improvement. Notably, Our KAT-T achieves the highest accuracy at 74.6%, outperforming GELU by 1.9%. This suggests that GR-KAN, as used in KAT-T, can significantly enhance the expressiveness of MLPs in vision transformers.

In addition to accuracy, we analyzed the computational cost of different activations by measuring throughput and peak memory usage on an NVIDIA A5000 GPU (Table 7). All activations had similar peak memory usage. However, KAT-T showed slightly lower throughput compared to baseline activations like ReLU, GELU, and SiLU. This suggests a trade-off between accuracy improvement and computational efficiency due to the increased complexity of rational function computations.

Table 7: Throughput and Peak memory for different activations on A5000 GPU. Input size is fixed to $[64, 1000, 512]$.

| Activation | ReLU | GeLU | SiLU | PReLU | Ours |
|---|---|---|---|---|---|
| **Throughput** (batch/s) | 2654 | 2643 | 2668 | 2644 | 2313 |
| **Peak Memory** (M) | 1380 | 1380 | 1380 | 1380 | 1380 |

Table 8: Ablation on rational function initialization, with KAT-T.

| Rational 1 Init. | Rational 2 Init. | IN-1k Top-1 |
|---|---|---|
| Random | Random | 53.2 |
| Identity | Identity | 69.7 |
| Swish | Swish | 74.4 |
| Identity | GeLU | 74.5 |
| Identity | Swish | **74.6** |

**Initialization.** We evaluate the KAT-T model using different initializations when training from scratch. 'Random' refers to initializing all $a_m, b_n$ values using Xavier initialization, without our variance-preserving init. As shown in Table 8, the "Identity - Swish" initialization achieves the best performance, which we have adopted as our default setting. 'Random' initialization results in significantly lower accuracy.

## 6 CONCLUSION

In this work, we introduced the Kolmogorov–Arnold Transformer (KAT), a novel architecture that successfully integrates Kolmogorov-Arnold Networks (KANs) into transformers, addressing key challenges associated with large-scale training scenarios. Our proposed Group-Rational KAN (GR-KAN) variant, with its rational activation functions, group-based parameter sharing, and variance-preserving initialization, demonstrated significant improvements in computational efficiency and scalability. Through extensive experiments on vision tasks, including image recognition, object detection, and semantic segmentation, KAT outperformed traditional MLP-based transformers, achieving superior accuracy on ImageNet1K while maintaining comparable computational demands.

ACKNOWLEDGMENT

This project is supported by the Ministry of Education, Singapore, under its Academic Research Fund Tier 2 (Award Number: MOE-T2EP20122-0006). We would like to acknowledge that computational work involved in this research work is partially supported by NUS IT's Research Computing group using grant numbers NUSREC-HPC-00001. We thank Weihao Yu, Qiuhong Shen and Runpeng Yu for valuable discussions.

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

## A  APPLICATION ON DOMAIN BEYOND VISION

In addition to the vision task we presented in the paper, in this appendix, we also provide experiments on applying KAT to tabular data and graph data.

**Tabular Experiment.** We choose TabTransformer (Huang et al., 2020) as our baseline for tabular classification. We test 15 publicly available binary classification datasets from UCI dataset (Bache & Lichman, 2013), AutoML challenge (Guyon et al., 2019) and Kaggle. We keep all experiment setup in the paper, but only modify architcture by replacing MLPs to GR-KANs.

The results are presented in Table 9. Our KAT, when integrated with TabTransformer, improves the AUC score by an average of 0.9. This demonstrates the potential of KAT in enhancing the performance of tabular data models.

Table 9: Comparison between KAT and the baseline TabTransformers on 15 datasets. The evaluation metric is AUC in percentage.

| Dataset | TabTransformer | TabTransformer+KAT |
|---|---|---|
| albert | 75.7 | 76.9**(+1.2)** |
| 1995_income | 90.6 | 91.1**(+0.5)** |
| dota2games | 63.3 | 64.1**(+0.8)** |
| hcdr_main | 75.1 | 76.2 **(+1.1)** |
| adult | 73.7 | 75.3 **(+1.6)** |
| bank_marketing | 93.4 | 93.8 **(+0.4)** |
| blastchar | 83.5 | 83.1**(-0.4)** |
| insurance_co | 74.4 | 75.9**(+1.5)** |
| jasmine | 85.3 | 85.6**(+0.3)** |
| online_shoppers | 92.7 | 93.2**(+0.5)** |
| philippine | 83.4 | 84.3 **(+0.9)** |
| qsar_bio | 91.8 | 92.6 **(+0.8)** |
| seismicbumps | 75.1 | 77.8**(+2.7)** |
| shrutime | 85.6 | 86.4**(+0.8)** |
| spambase | 98.5 | 98.6**(+0.1)** |
| Average | 82.8 | 83.7 **(+0.9)** |

**Graph Experiments.** We apply our methods on top of Graph Transformer (Dwivedi & Bresson, 2020) and see if it improves the performance. We replicate its experiments on ZINC (Irwin et al., 2012) for graph regression and PATTERN (Abbe, 2017) and CLUSTER (Dwivedi et al., 2023) for node classification. We keep all setup the same as its original paper and only modify the MLP layers. The results are shown in Table 10. KAT, again, demonstrates improved performance on the graph datasets.

Table 10: Comparison of scores graph dataset against the Graph Transformer (Dwivedi & Bresson, 2020) baseline with 500k model parameters.

| Model | ZINC | CLUSTER | PATTERN |
|---|---|---|---|
| Graph-Transformer | $0.226 \pm 0.014$ | $73.169 \pm 0.622$ | $84.808 \pm 0.068$ |
| Graph-Transformer + KAT | $\mathbf{0.211} \pm 0.013$ | $\mathbf{74.089} \pm 0.600$ | $\mathbf{85.293} \pm 0.065$ |

## B  GR-KAN OUT OF TRANSFORMER

Our proposed GR-KAN can be applied outside the scope of transformer. In this section we apply it on two tasks. One on simple function fitting and the other on solving partial differential equation.

**Function fitting.** In our function fitting experiments, we selected functions based on examples presented in the KAN paper. We adopted a simple neural network architecture consisting of three layers: an input layer with 2 neurons, a hidden layer with 5 neurons, and an output layer with 1 neuron, denoted as a $[2 \rightarrow 5 \rightarrow 1]$ structure. The models were trained for 1000 epochs using the Adam optimizer with a learning rate of 0.001.

The performance of each method is evaluated using the mean squared error (MSE), where lower values indicate better function approximation. We compare the results for three models: a standard Multi-Layer Perceptron (MLP), the KAN model, and our proposed GR-KAN model. The MSE values for each function are presented in Table 11.

Table 11: Mean Squared Error (MSE) for Function Fitting. Lower values indicate better performance.

| Method | $\exp\{\sin(x^2 + y^2)\}$ | $\exp\{\sin(\pi x) + y^2\}$ | $\exp\{J_0(20x) + x_2^2\}$ | $xy$ | $\frac{x}{y}$ | $(x + y) + xy$ |
|---|---|---|---|---|---|---|
| MLP | 0.4307 | 180.3786 | 43.4192 | **80.8309** | 0.0766 | 0.0503 |
| KAN | 0.6618 | 403.9234 | 194.7122 | 83.8479 | 1.8517 | 4.5709 |
| GR-KAN | **0.0034** | **19.3789** | **20.0403** | 90.5357 | **0.0016** | **0.0221** |

The results clearly show that our GR-KAN model achieves significantly lower MSE values in most cases compared to both the MLP and KAN models. This demonstrates the improved ability of GR-KAN to fit complex functions effectively, highlighting its robustness and accuracy in regression tasks.

**PDE Solving.** For solving the PDE, we focus on a one-dimensional damped harmonic oscillator. This system is governed by the equation:

$$m\frac{d^2 u}{dt^2} + \mu\frac{du}{dt} + ku = 0, \tag{18}$$

where $m$ represents the mass, set to 1 in our experiments. The term $\mu$ denotes the damping coefficient, while $k$ corresponds to the stiffness constant. We define the initial conditions as $u(0) = 1$ and $u'(0) = 0$, providing a complete setup for the problem.

The exact analytical solution of this oscillator can be expressed as:

$$u(t) = e^{-d \cdot t}\left(A\cos(\omega t + \phi)\right),$$

where $d = \mu/2w_0 = \sqrt{k}, w = \sqrt{w_0^2 - d^2}, \phi = \arctan(-d/w)$ are derived from the parameters of the equation and the initial conditions. To approximate this solution, we use three methods: a standard Multi-Layer Perceptron (MLP), the KAN model, and our proposed GR-KAN model.

Our experimental results, presented in Table 12, compare the performance of different methods based on the L2 error and training loss between predicted and true solutions over a specified time interval.

The KAN model achieves the best performance but requires more parameters and significantly longer training time ( 20 minutes). In contrast, the GR-KAN model trains much faster (4 minutes) while outperforming the pure MLP in solving this type of PDE problem.

Table 12: Comparison of L2 Loss and Predictions for MLP and KAN

| Type | L2 error (last step) | Prediction | Loss Curve |
|------|---------------------|------------|------------|
| | | $w_0 = 10$ | |
| MLP (GELU) | $2.0216 \times 10^{-4}$ | | |
| GR-KAN | $6.3909 \times 10^{-6}$ | | |
| KAN | $1.6125 \times 10^{-8}$ | | |
| | | $w_0 = 50$ | |
| MLP (GELU) | $1.1805 \times 10^{-1}$ | | |
| GR-KAN | $5.2515 \times 10^{-2}$ | | |
| KAN | $3.7762 \times 10^{-2}$ | | |

## C  DETAILED ABLATION STUDY

We conducted a detailed ablation study to evaluate the contributions of three key components and parameter grouping in KAN. These experiments were performed using the KAT-Tiny model on ImageNet, training for 300 epochs. Specifically, we analyzed:

- **Base Function**: Replacing the rational base function with the B-spline used in KAN.
- **Group-wise Computation**: Replacing group-wise weight sharing with distinct parameters for each channel.
- **Initialization**: Replacing the proposed initialization with PyTorch's default random initialization.

The results of the ablation study are summarized in Table 13, including Top-1 accuracy, training time, and MACs

**Ablation 1: Base Function**   We replaced the rational base function with B-splines, implemented using the De Boor-Cox algorithm in PyTorch, as no CUDA implementation is available.

The results show that the base function slightly impacts performance but significantly affects runtime. According to `Exp 3` and `Exp 6`, the rational function provides a small performance improvement over B-splines. Although the MACs differ only slightly, the PyTorch-based B-spline implementation is much slower than the CUDA-optimized rational function.

**Ablation 2: Group-wise Computation**   We replaced group-wise weight sharing with distinct parameters for each channel.

As shown in `Exp 4` and `Exp 6`, group-wise computation is critical for efficiency. While it results in a minor accuracy drop from 74.8% to 74.6%, it reduces training time drastically from 38 hours to 12 hours, making it the most significant factor for efficiency improvements.

**Ablation 3: Initialization**   We replaced the proposed initialization with PyTorch's default initialization.

As shown in `Exp 6` and `Exp 5`, proper initialization is crucial for achieving fast and reliable convergence and improving performance, particularly for rational functions with higher-order terms. Without proper initialization, terms of different orders are initialized at similar scales, leading to instability. This issue is more severe for rational functions compared to B-splines, as highlighted by the comparison between `Exp 2` and `Exp 5`.

**KAN with Parameter Grouping**   In `Exp 2`, we explored parameter sharing in KAN by applying parameter grouping to the original ViT+KAN architecture. This modification significantly improved the training speed, reducing the time from 43 hours to 20 hours. However, it resulted in a performance drop of 2.7%.

| Exp ID | Rational | Group | Initialization | Top-1 (%) | Train Time | MAC |
|:---:|:---:|:---:|:---:|:---:|:---:|:---:|
| 1 | ✗ | ✗ | ✗ | 64.9 | 43h | 1.78G |
| 2 | ✗ | ✓ | ✗ | 62.2 | 20h | 1.15G |
| 3 | ✗ | ✓ | ✓ | 73.0 | 20h | 1.15G |
| 4 | ✓ | ✗ | ✓ | **74.8** | 38h | 1.76G |
| 5 | ✓ | ✓ | ✗ | 53.2 | **12h** | **1.13G** |
| 6 | ✓ | ✓ | ✓ | **74.6** | **12h** | **1.13G** |

Table 13: Ablation study on KAT-Tiny with different components.

## D  VARIANCE ANALYSIS OF THE KAN LAYERS

In this section, we discuss why the original initialization in KAN fails to maintain activation variance across layers.

Specifically, the B-spline coefficients $c_i$ are initialized as $\mathcal{N}(0, \sigma^2)$ with $\sigma = 0.1$, and $w_s = 1$ and $w_b \sim U[-\frac{6}{\sqrt{d_{in}+d_{out}}}, \frac{6}{\sqrt{d_{in}+d_{out}}}]$ are initialized according to the Xavier initialization (Glorot & Bengio, 2010). The combined output variance of the model can be expressed as:

$$Var[\phi(x)] = Var[w_b\texttt{silu}(x)] + Var[w_s\texttt{spline}(x)] = 3\mathbb{E}[\texttt{silu}^2(x)] + \mathbb{E}[\texttt{spline}^2(x)] \quad (19)$$

**Zero-th Order Case**. If we assume the input $x$ is normally distributed, $x \sim \mathcal{N}(0, \sigma_x^2)$ and consider a zero-th order spline, the variance of $\texttt{spline}(x)$ at any point $x$ is simply:

$$\mathbb{E}[\texttt{spline}^2(x)] = \sum_i c_i^2 Var[B_i(x)] = \sigma^2 \sum_i Var[B_i(x)] = \sigma^2 = 0.01 \quad (20)$$

For the SiLU activation function, although exact variance calculations are complex, numerical estimations indicates $\mathbb{E}[\texttt{silu}^2(x)] \approx 0.355\sigma_x^2$. Combining these, we find $Var[\phi(x)] \approx 0.01 + 1.064\sigma_x^2 \neq Var[x]$.

This indicates that, under zero-th order spline, $Var[\phi(x)] \neq Var[x]$.

**Higer Order Case**. With higher-order splines, the variance instability might increase. This because, increasing the order of a spline leads to excessive smoothing. This smoothing effect reduces the variation in the function values, causing $Var[\phi(x)]$ to become smaller.

The B-spline is defined recursively as:

$$B_{i,p}(t) = \frac{t - t_i}{t_{i+p} - t_i}B_{i,p-1}(t) + \frac{t_{i+p+1} - t}{t_{i+p+1} - t_{i+1}}B_{i+1,p-1}(t)$$

shows that each higher-order basis function $B_{i,p}(t)$ a weighted summation of a weighted average of two lower-order basis functions $B_{i,p-1}(t)$ and $B_{i+1,p-1}(t)$. as the order $p$ increases, the B-spline becomes wider and smoother.

In the extreme case, as $p \to \infty$, the smoothing effect causes the basis functions to become nearly uniform across the domain, i.e., $B_{i,p}(t) \approx C$, a constant. Consequently, as, when $p \to \infty$, the variance of the output converges to zero, $Var[\phi(x)] \to 0$. This excessive smoothing leads to instability in the activation variance, effectively flattening all variations. Regardless of weight initialization, the function becomes unstable for use in neural networks.

Based on all analysis above, the default initialization opposes the essential variance-preserving principle.

## E  EXPLICIT GRADIENT CALCULATION

n our CUDA implementation, we compute the gradients $\frac{\delta F}{\delta a_m}$, $\frac{\delta F}{\delta b_n}$ and $\frac{\delta F}{\delta x}$ using the following approach:

$$\frac{\delta F}{\delta a_m} = \frac{x^m}{Q(x)}, \quad \frac{\delta F}{\delta b_n} = ax^n \frac{A(x)}{|A(x)|}\frac{P(x)}{Q(x)^2}, \quad \text{and} \quad \frac{\delta F}{\delta x} = \frac{\delta P(x)}{\delta x}\frac{1}{Q(x)} - \frac{\delta Q(x)}{\delta x}\frac{P(x)}{Q^2(x)} \quad (21)$$

where $A(x) = b_1 x + \cdots + b_n x^n$, $\frac{\delta P(x)}{\delta x} = a_1 + 2a_2 x + ma_m x^{m-1}$ and $\frac{\delta Q(x)}{x} = \frac{A(x)}{|A(x)|}(b_1 + 2b_2 x + nb_n x^{n-1})$.

## F  INITIALIZING $a_m, b_n$

Given a ground-truth function $g(\cdot)$ and a parameterized rational function $F(\cdot; \{a_m\}, \{b_n\})$, we run a linear least square to determine $\{a_m\}, \{b_n\}$. Specifically, we optimize the following function:

$$\min_{\{a_m\}, \{b_n\}} \frac{1}{2}\sum_{i=1}^{N}(g(x_i) - F(x_i; \{a_m\}, \{b_n\}))^2$$

We uniformly sample 1000 points $x_i$ from the interval $[-3, 3]$. $\{a_m\}, \{b_n\}$ are randomly initialized. In practice, we solve this using the Levenberg-Marquardt algorithm, available in the MINPACK package.

In the end, this is easily done by calling `scipy.optimize.curve_fit`.

## G  WHY USING RATIONAL FUNCTION?

In this section, we explain why we chose rational functions as our base function. The main reasons are efficiency, prior successful use, and strong theoretical properties.

**Efficiency Perspective.** Evaluating polynomials involves simple operations that are well-suited for parallel computing. This makes rational functions computationally efficient for large-scale models.

**Practical Perspective.** Rational activations have already been successfully used as activation functions in neural networks (Boullé et al., 2020; Molina et al., 2020).

**Theoretical Advantage.** Rational functions can approximate a wider range of functions—including those with singularities or sharp variations—more efficiently and accurately than polynomials (Walsh, 1935; Baker Jr & Gammel, 1961). Since B-splines are essentially sums of local polynomials, rational functions offer a theoretical advantage over B-splines for modeling complex behaviors.

Given these reasons, we adopt rational functions as the base functions in our KAN layers to enhance the model's expressiveness, stability, and computational efficiency.

## H  DERIVATION AND CALCULATION OF FLOPS

Given the function:

$$F(x) = \frac{a_0 + a_1 x + \cdots + a_m x^m}{1 + |b_1 x + \cdots + b_n x^n|}$$

### H.1  PLAIN COMPUTATION

NUMERATOR

The numerator is a polynomial of degree $m$:

- Multiplications: There are $\frac{m(m-1)}{2}$ multiplications for computing powers of $x$ and $m$ multiplications for coefficients $a_i$, giving $\frac{m(m-1)}{2} + m$.
- Additions: There are $m$ additions to sum up the polynomial terms.

DENOMINATOR

The denominator involves the absolute value of a polynomial of degree $n$:

- Multiplications: There are $\frac{n(n-1)}{2}$ multiplications for powers of $x$ and $n$ multiplications for coefficients $b_i$, giving $\frac{n(n-1)}{2} + n$.
- Additions: There are $n$ additions for polynomial terms and 1 additional addition after the absolute value operation.
- Absolute value operation: 1 absolute value calculation.

**Division.** There is 1 division operation for the final computation of $F(x)$.

**Total FLOPs.** The total FLOPs for any $m$ and $n$ are:

Multiplications: $\frac{m(m-1)}{2} + \frac{n(n-1)}{2} + m + n + 1$,   Additions: $m+n+1$,   Absolute Value: 1,   Division: 1

In case $m = 5$ and $n = 4$, there are totally 16 multiplications, 10 summations, 1 absolute value and 1 division. In total 28.

### H.2  HORNER'S METHOD

Using horner's method, for a polynomial of order $m$, we need $m$ summations and $m$ multiplications.

Thus, for numerator, we need $m$ summations and $m$ multiplications. For denominator, we need $n + 1$ summations and $n$ multiplications. In total, we need $m + n + 1$ summation, $m + n$ multiplications, 1 absolute value, and 1 division.

In case $m = 5$ and $n = 4$, there are a total of 21 FLOPs, comprising 9 multiplications, 10 summations, 1 absolute value, and 1 division.

## I    HYPER-PARAMETERS FOR KAT MODEL

The hyper-parameter for training KAT model on ImageNet-1k is shown in Table 14.

Table 14: Hyper-parameters of KAT on ImageNet image classification.

| | **KAT** | | |
|---|---|---|---|
| | **Tiny** | **Small** | **Base** |
| Input resolution | | $224^2$ | |
| Epochs | | 300 | |
| Batch size | | 1024 | |
| Optimizer | | AdamW | |
| Adam $\epsilon$ | | $1 \times 10^{-8}$ | |
| Adam $(\beta_1, \beta_2)$ | | (0.9, 0.999) | |
| Learning rate | | $1 \times 10^{-3}$ | |
| Learning rate decay | | Cosine | |
| Gradient clipping | | None | |
| Warmup epochs | | 5 | |
| Weight decay | | 0.05 | |
| Rand Augment | | 9/0.5 | |
| Repeated Augmentation | | off | |
| Cutmix | | 1.0 | |
| Mixup | | 0.8 | |
| Cutmix-Mixup switch prob | | 0.5 | |
| Random erasing prob | | 0.25 | |
| Label smoothing | | 0.1 | |
| Peak stochastic depth rate | 0.1 | 0.1 | 0.4 |
| Random erasing prob | | 0.25 | |
| EMA decay rate | | 0.9999 | |

**Model Variant.** We select the configurations of KAT to be identical with those used in ViT (Dosovitskiy et al., 2021), as summarized in Table 15. All variants use an input patch size of $16 \times 16$.

| **Model** | **Layers** | **Hidden Size D** | **MLP Size** | **Heads** | **Params** |
|---|---|---|---|---|---|
| KAT-Tiny | 12 | 192 | 768 | 3 | 5.7M |
| KAT-Small | 12 | 384 | 1536 | 6 | 22.1M |
| KAT-Base | 12 | 768 | 3072 | 12 | 86.6M |

Table 15: Details of KAT model variants.

### I.1    ABLATION AND ANALYSIS

This section outlines the key hyperparameter settings used for KAT, determined through ablation studies and preliminary experiments.

**Number of Groups**. We conducted an ablation study with the KAT-Tiny model to determine the optimal number of groups. The results showed that accuracy improved slightly up to 8 groups, with no further gains beyond that. Based on these findings, we set the number of groups to **8**, which balances simplicity and performance.

Table 16: Impact of group numbers on KAT-Tiny Top-1 accuracy.

| Group Number | 2 | 4 | 8 | 16 | 32 |
|---|---|---|---|---|---|
| **KAT-Tiny Top-1 Accuracy** | 74.2 | 74.3 | 74.6 | 74.7 | 74.6 |

**Maximum Order of Rational Function**. We set the rational order to $(m = 5, n = 4)$ in the paper, following the default configuration in the PAU paper. We conduct a ablation experiment with the KAT-Tiny model to confirm this choice

Table 17: Impact of rational order on KAT-Tiny Top-1 accuracy.

| Order (m, n) | (3, 2) | (5, 4) | (7, 6) |
|---|---|---|---|
| **KAT-Tiny Top-1 Accuracy** | 74.2 | 74.6 | 74.6 |

The experiment show that increasing the order beyond $(m = 5, n = 4)$ provides no additional accuracy gains, making it a practical and efficient choice.

**Benefit of CUDA Implementation.** To evaluate the efficiency improvements introduced by our CUDA implementation discussed in Section 4.2, we conducted experiments to measure both forward pass speed and peak memory usage. Specifically, we compared our CUDA implementation against two alternative methods. The first is called *Torch Looped*, which loops over each channel group, applies the rational function, and then concatenates the results. The second is called *Torch Vectorized*. In this method, the input tensor is reshaped according to the channel groups, the rational function is applied in a vectorized manner, and the tensor is reshaped back to its original form. We compare these three implementation on A5000 GPU, under 1) different group number $g \in \{1, 2, 4, 8, 16\}$. 2) different input dim $D \in \{128, 256, 512, 1024, 2048\}$

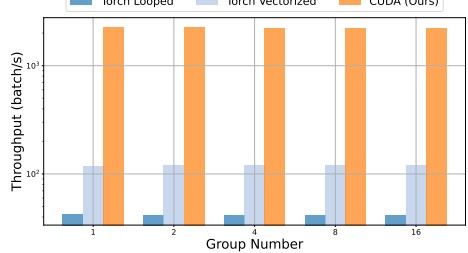

(a) Throughput (batch/s) for Different Group Sizes. Larger the better.

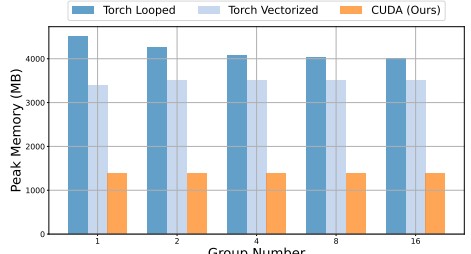

(b) Peak Memory (MB) for Different Group Sizes. Smaller the better.

Figure 6: Comparison of Throughput and Peak Memory for Different Methods and Group Sizes. Input size is fixed to $[64, 1000, 512]$.

The results, presented in Figure 6 and Figure 7, clearly demonstrate that our CUDA implementation significantly outperforms both the Torch Looped and Torch Vectorized implementations, offering superior speed and memory efficiency.

**Visualization of Trained Functions** An important aspect to examine is the behavior of the trained rational functions. As shown in Figure 8, we plot the functions for KAT-S with $g = 8$ across all 12 layers. The results indicate that within each layer, the rational functions exhibit similar trends, while the functions across different layers tend to differ from each other. Another interesting finding is that most of the fitted rational functions are (near)-identity. This indicates that the model learns to perform relatively simple operations rather than overly complex ones.

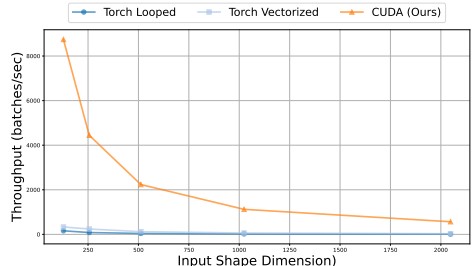 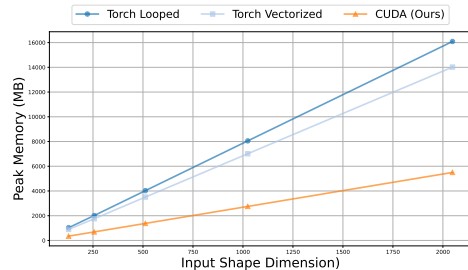

(a) Throughput (batch/s) for Input Dimension Sizes. Larger the better.

(b) Peak Memory (MB) for Input Dimension Sizes. Smaller the better.

Figure 7: Comparison of Throughput and Peak Memory for Different Methods and Input Dimension Sizes. Group size is fixed to 8.

## J  DISCUSSION AND FUTURE WORK

**Discussion.** Our study highlights KAT's potential as a good alternative to MLP-based transformers, especially in large-scale vision tasks. This integration introduces exciting opportunities for broad applications. For example, employing KAT architectures might help development of language models.

However, KAT is not without its challenges. A primary concern is running speed. Even with the CUDA optimized code, the rational function is still slower than plain activation like ReLU and GELU. Another issue is the stability when using rational functions in neural networks. The higher order gradients for $a_m$ and $b_n$ can become unstable because of their dependence on the input power. Integrating these functions into the backpropagation process could introduce complications.

Additionally, it is important to acknowledge that our GR-KAN represents a hybrid model. On the one hand, GR-KAN is a KAN layer with shared edges and a rational base function. On the other hand, it can be interpret as MLP with a redesigned activation placed before the linear layer. It leverages the computational simplicity of MLPs but maintains some characteristics of KANs. However, GR-KAN is not a pure KAN model. Instead, it merges advantages from both systems to enhance overall functionality.

**Future Work.** There are multiple directions of KAT for future research. One potential area of exploration is to find alternative base functions to further improve computational efficiency and compatibility with emerging hardware architectures. Currently, rational functions serve as one option, but other possibilities exist. These include Fourier transformations (Noesis, 2024), Wavelet transforms (Bozorgasl & Chen, 2024b), and Gaussian radial bases (Li, 2024).

Additionally, expanding the applicability of KAT to other domains beyond vision tasks, such as natural language processing or reinforcement learning, could unlock new opportunities for performance gains. Further research could also investigate hybrid models (Yang et al., 2022; Yu et al., 2023), or adaptive mechanisms for dynamically selecting between KAN and MLP layers based on the complexity of the task, thereby optimizing resource utilization. Finally, addressing the remaining scalability challenges, particularly in terms of memory footprint and inference speed, will be crucial for deploying KAT in real-world applications at scale.

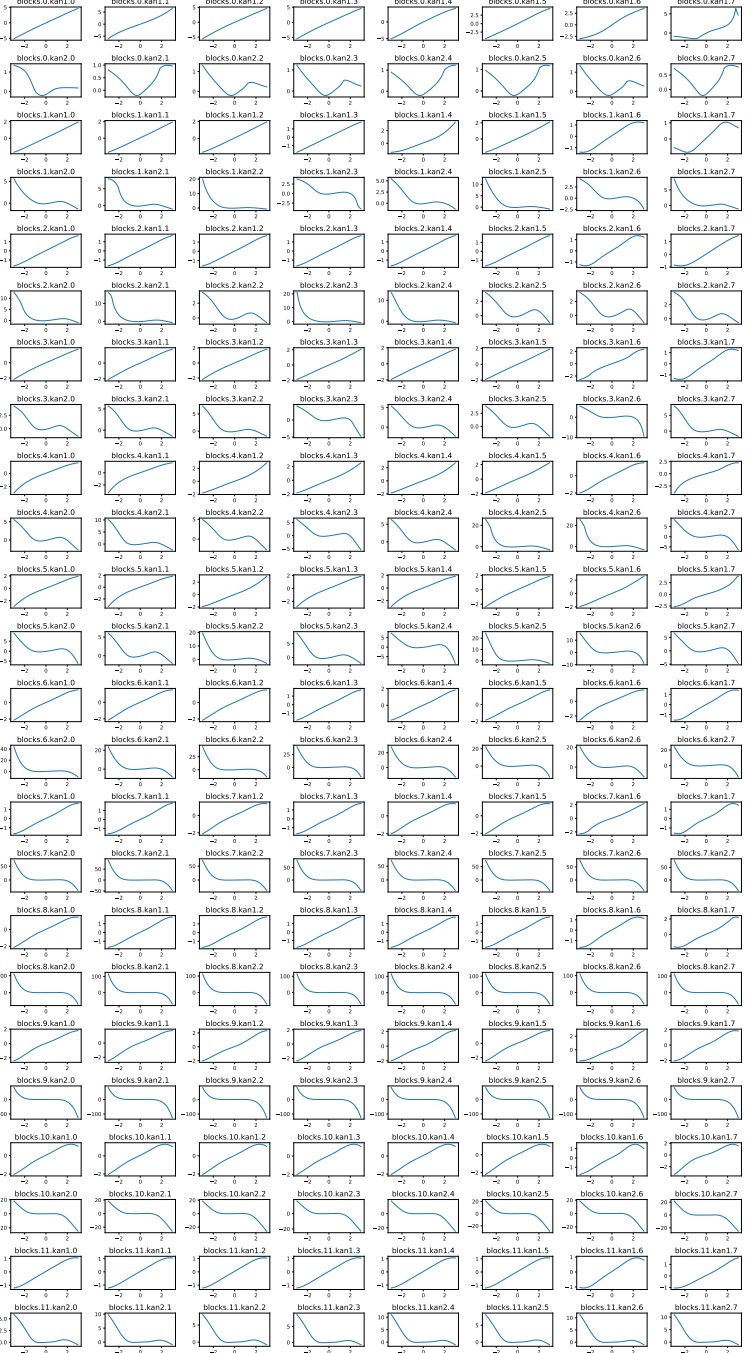

Figure 8: Fitted rational functions for KAT-S model, with 12 layers and 8 groups.

