# OpenReview forum: "Kolmogorov-Arnold Transformer"
_ICLR.cc/2025/Conference — ICLR 2025 Poster_

### Official Review · Reviewer_s19p · 2024-10-31

**Soundness:** 4
**Presentation:** 3
**Contribution:** 4
**Rating:** 6
**Confidence:** 4

**Summary:**

The author proposes and implements Kolmogorov-Arnold Transformer (KAN) architecture. Especially, they propose rational activation functions, group KAN, and variance-preserving initialization, demonstrating improved performance over conventional vision transformers across various tasks.

**Strengths:**

* Comparative experiments are conducted on image classification, object detection, instance segmentation, and semantic segmentation.
* Challenges and solutions are clearly explained, supported by experimental results.
* The proposed method is sufficiently general to be applicable across different transformer architectures.

**Weaknesses:**

* In Tab 1, given the efficient CUDA implementation, experimental results of GPU computation are needed. Specifically, the inference time on a specified GPU should be provided.
* The choice of the number of groups in group-rational KAN is not discussed. What is the reason for the current choice of group number? An experiment analyzing the effect of different group numbers would be beneficial for better understanding.
* For consistency, it would be helpful to include the GPU information used for image classification task as well.

**Questions:**

* Could you specify where in the report it states that KAN is "10x slower than MLPs, given the same number of parameters"?

---

> ### Author Response · Authors · 2024-11-21
> **Thank Reviewer s19p for the suggestions**
>
> We thank R-s19p for the nice suggestions.
>
> `>>> Q1` **GPU running time**
>
> `>>> A1` That is a great question. In the paper, we report the CUDA running time on an A5000 GPU in `Appendix J.1`, for each layer with different width and group number. Our implementation is faster than the pure PyTorch version.
>
> Additionally, as requested, here is the full model throughput on an A5000 GPU using FP32. While our model is slightly slower than a pure ViT, it is significantly faster than ViT+KAN.
>
> |Model|Throughput (images/s)|
> |--|--|
> |ViT-Ti/16| 1102 |
> |ViT-T + KAN| 321 |
> |KAT-T| 934 |
>
> `>>> Q2` **Group Number**
>
> `>>> A2` We thank the reviewer the the nice question. To verify this, we conducted an ablation study using the KAT-Tiny model to explore the impact of different group numbers.
>
> Our results showed that increasing the number of groups improved accuracy slightly up to 8 groups, with no further gains beyond that. Based on these findings, we chose **8 groups** as the optimal configuration, providing a good trade-off between simplicity and performance.
>
> |Group Number | 2 | 4 | 8 | 16 | 32|
> |--|--|--|--|--|--|
> | KAT-Tiny Top-1|74.2 | 74.3| 74.6 | 74.7 | 74.6|
>
> The results has been added to revision `Appendix J.1`.
>
> `>>> Q3` **GPU information**
>
> `>>> A3` For the image classification tasks, some experiments were conducted on servers with 8×A5000 GPUs. For the larger KAT-B model, we used servers equipped with 8×H100 GPUs.
>
> `>>> Q4` **KAN statement**
>
> `>>> A4` Thanks for the question. The authors of KAN made this statement in its paper, at the section `Final takeaway: Should I use KANs or MLPs?`. They point out that
>
> > Currently, the biggest bottleneck of KANs lies in its slow training. KANs are usually 10x slower than MLPs, given the same number of parameters.

---

> ### Comment · Reviewer_s19p · 2024-11-25
>
> I appreciate that the authors incorporated a study on GPU computation and the group number of KAT. All of my primary concerns have been resolved.

---

> ### Author Response · Authors · 2024-11-25
>
> Thank R-s19p so much! We’re glad the concerns have been resolved and appreciate the valuable feedback.

---

### Official Review · Reviewer_DyMH · 2024-11-01

**Soundness:** 3
**Presentation:** 3
**Contribution:** 3
**Rating:** 6
**Confidence:** 4

**Summary:**

In this work, authors propose Kolgomorov-Arnold Transformer (KAT) as an improvement to standard transformer architecture, which uses MLP networks, by replacing MLPs with KAN networks. Authors propose a number of improvement over vanilla KAN architectures to make them scalable in practical settings and achieve better results: 1.) they swap b-spline functions with rational function for speed 2.) they make use of parameters sharing (grouping) 3.) they employ a variance preserving initialization.
Results show that KAT achieves better results than ViT on certain benchmarks.

**Strengths:**

- The topic is highly relevant, as literature has not focused much on improving the MLP aspect of transformers
- KANs have emerged as an interesting alternative to MLPs, however, they present issues (e.g. scaling) which make them unfavorable w.r.t. MLPs
- The proposed approach is scalable, faster, and requires less parameters than KANs, making it usable in practical settings and on large-scale datasets such as ImageNet-1k or MSCOCO
- The experimental evaluation comprises different tasks such as classification, object detection and segmentation.

**Weaknesses:**

Here are my doubts about this work:
- While the results for ImageNet are convincing, the proposed KAT still lags behind traditional architectures such as ConvNext or even transformers such as SwinB in segmentation.
- A proper ablation study is missing (e.g. combinations of KAN + rational function + grouping + initialization). It is difficult to evaluate the contribution of each proposed change quantitatively. Also I wonder how simple KAN perform with parameter grouping in terms of MACS/FLOPS and results?
- It is not clear how you choose the hyperparameters for KAT such as number of groups and maximum order of rational function (m)
- (More of a philosophical doubt) KAN are based on the Kolmogorov-Arnold representation theorem, which states that a multivariate function can be approximated by a composition of univariate functions, while your proposed method only uses one irrational function. Can you still say it's based on the Kolmogorov-Arnold theorem?
- (minor) some typos and missing notation are present in the paper (see questions below)

**Questions:**

- In Eq. 5, 6, and 7 what are MSA and LN?
- Missing text in paragraph name in line 485
- In Tab. 4 KATDet-S is reported as best (41.5) but ConvNext and SwinT are higher (41.7 and 41.6)
- I feel like the change you propose can be applied to KANs also outside of the transformer architecture. Perhaps it could be useful to add a comparison between KANs and GR-KANs on toy problems

---

> ### Author Response · Authors · 2024-11-21
> **Response to Reviewer DyMH (Part I)**
>
> We sincerely thank Reviewer DyMH for the thoughtful comments and suggestions. We have conducted new experiments to address the questions raised.
>
> `>>> Q1` **Comparision with ConvNext and Swin**
>
> `>>> A1` Great question. While our performance is behind ConvNext and Swin, this is due to **differences in micro-architecture**. The hierarchical designs in ConvNext and Swin are inherently more effective than the plain ViT-style architecture we used, especially for tasks like segmentation.
>
> **New Results**: Despite this, our method can be extended to hierarchical architectures. We tested replacing MLP layers in Swin and $1 \times 1$ convolutions in ConvNext with GR-KAN layers.
>
> Results on ADE20K show that our GR-KAN consistently outperforms the original MLP layers across these architectures.
>
> |Model|mIOU(\%)|
> |--|--|
> |Swin-T|45.8|
> |Swin-T+GR-KAN|**46.6**|
> |ConvNext|46.7|
> |ConvNext+GR-KAN|**47.3**|
>
> Note that, Our primary goal was to demonstrate that KAN outperforms MLP within transformers. To focus on this comparison, we intentionally adopted a simple ViT-like design, as noted on `Line 399`.
>
> `>>> Q2` **Detailed ablation study**
>
> `>>> A2` We truly thank the suggestions from R-DyMH. As suggested, we conducted a detailed ablation study focusing on three components and the `KAN with parameter grouping modification`. These experiments were performed on the KAT-Tiny model using ImageNet with 300 epochs of training.
>
> We analyzed the effects of removing the following components:
> 1) Rational base function, replaced with B-splines as used in KAN.
> 2) Group-wise weight sharing, replaced with distinct parameters for each channel.
> 3) Proper initialization, replaced with random initialization.
>
> **Ablation 1: Base function**:  We replaced the rational base function with a B-spline implemented in torch using the De Boor-Cox algorithm due to the lack of a pure CUDA implementation for B-splines.
>
> - **Observation**: The choice of base function had a minor impact on performance and influenced runtime significantly.
> - **Key Insight**: As shown in `Exp 3` and `Exp 6`, using rational functions slightly improved performance over B-splines. While the increase in MAC count was minimal, our pure CUDA implementation for rational functions ran considerably faster than the torch-based B-spline implementation.
>
>
> **Ablation 2: Group-wise computation**: We replaced group-wise weight sharing with a setup where each channel had distinct parameters.
>
> - **Observation**: Group-wise computation proved to be the most significant factor for efficiency.
> - **Key Insight**: According to `Exp 4` and `Exp 6`, group-wise computation reduced training time from 38 hours to 12 hours, with a marginal drop in accuracy from 74.8% to 74.6%.
>
>
> **Ablation 3: Initialization**: We replaced proper initialization with the default initialization in `torch`.
>
> - **Observation**: Proper initialization was critical for fast and reliable convergence, particularly for rational functions with higher-order terms.
> - **Key Insight**: As shown in `Exp 6` and `Exp 5`, without proper initialization, terms of different orders were initialized at similar scales, causing instability. This issue was more pronounced for rational functions than for B-splines, as evidenced by the performance differences between `Exp 2` and `Exp 5`.
>
>
>
> **KAN with parameter grouping**: In `Exp 2`, we explored parameter sharing in KAN by applying parameter grouping to the original ViT+KAN architecture. This modification significantly improved the training speed, reducing the time from 43 hours to 20 hours. However, it resulted in a performance drop of 2.7%.
>
> We have incorporated the results in `Appendix C`.
>
> |Exp ID| Rational | Group | Initiation | Top-1| Train Time | MAC |
> |--|--|--|--|--|--|--|
> |1| &#10008; | &#10008; | &#10008; | 64.9| 43h | 1.78G |
> |2| &#10008; | &#10004; | &#10008; | 62.2 | 20h | 1.15G |
> |3| &#10008; | &#10004; | &#10004; | 73.0 | 20h | 1.15G |
> |4| &#10004; | &#10008; | &#10004; | **74.8** | 38h | 1.76G |
> |5| &#10004; | &#10004; | &#10008; | 53.2| **12h** |**1.13G**|
> |6| &#10004; | &#10004; | &#10004; | **74.6**| **12h** | **1.13G**|

---

> ### Author Response · Authors · 2024-11-21
> **Response to Reviewer DyMH (Part II)**
>
> `>>> Q3` **Hyperparameters for KAT**
>
> `>>> A3` Thanks for the question.
>
> - **Number of groups**: As suggested, we conducted an ablation study with the KAT-Tiny model to determine the optimal number of groups.
>
>     The results showed that accuracy improved slightly up to 8 groups, with no further gains beyond that. Based on these findings, we chose to use 8 groups, which offer a good balance between simplicity and performance.
>
>     |Group Number | 2 | 4 | 8 | 16 | 32|
>     |--|--|--|--|--|--|
>     | KAT-Tiny Top-1|74.2 | 74.3| 74.6 | 74.7 | 74.6|
>
> - **Maximum order of rational**: We use the rational order
> (m=5,n=4), as it is the default setting in the PAU paper. We also conduct a preliminary experiment to confirm this choice
>
>     |Order Numer (m,n) | (3,2) | (5,4) | (7,6) |
>     |--|--|--|--|
>     | KAT-Tiny Top-1|74.2 | 74.6 | 74.6 |
>
>      The results indicate that increasing the order beyond
> (m=5,n=4) provides no additional accuracy gains. It stands as a practical and efficient choice.
>
> The results has been added to `Appendix J.1` in the revision.
>
> `>>> Q4` **Connecting to theorem**
>
> `>>> A4` Thanks. It is true that the KA theorem states that `a multivariate function can be approximated by a composition of univariate functions`.
>
> However, we are **not** using just a single univariate rational function. Instead, we approximate the multivariate function by summing several univariate rational functions. Some of these rational functions may share parameters, but the key idea remains consistent with the KA theorem.
>
> `>>> Q5` **Typos and missing notation**
>
> `>>> A5`
> - **MSA and LN**: MSA stands for multi-head self attention and LN stands for layer norm.
> - **Missing text**: We change the paragraph title from `v.s. Activation Function` to `GR-KAN v.s. Activation Function`.
> - **Annotation**: Sorry for the wrong annotation. We have revise the manuscript.
>
>
> `>>> Q6` **Toy example**
>
> `>>> A6` We truly appreciate the question. Our GR-KAN can indeed be applied to other tasks. In this answer we test on 2 tasks.
>
> - **Regression**: We first test on fitting the GR-KAN onto some special functions. The functions were selected based on the examples used in the KAN paper. We use the $[2\to5\to1]$ network architecture trained for 1000 epochs with the Adam optimizer and a learning rate of 0.001. The MSE results, shown below, indicate that smaller values are better. GR-KAN achieves the best performance.
>
>
>
> |Method | $\exp\{\sin(x^2 + y^2)\}$|$\exp\{\sin(\pi x) + y^2\}$ | $\exp\{J_0(20 x) + x^2_2\}$ | $xy$ |$\frac{x}{y}$ | $(x + y) + xy$
> |--|--|--|--|--|--|--|
> | MLP (ReLU) |0.4307 | 180.3786| 43.4192| 80.8309| 0.0766 0.0503 |
> KAN | 0.6618 | 403.9234 | 194.7122| 83.8479 | 1.8517 |4.5709
> GR-KAN |  0.0034 | 19.3789| 20.0403 | 90.5357 | 0.0016| 0.0221 |
>
> - **PDE solving**: To solve a PDE, we use a one-dimensional damped harmonic oscillator governed by:
>
> $$m \frac{{d^2 u}}{{dt^2}} + \mu \frac{{du}}{{dt}} + k u = 0$$
> where $m$ is the mass. $\mu$ is the damping coefficient. $k$ is the stiffness constant. With the initial condition $u(0) = 1,u'(0) = 0,m = 1$. The exact solution should be $u(t) = e^{-d \cdot t} \cdot (A \cos(\omega t + \phi))$, where $d = \mu / 2 w_0 = \sqrt{k}, w = \sqrt{w_0^2 - d^2}, \phi = \arctan(-d / w)$. We solve this using MLP, KAN, and GR-KAN with a network architecture of $[2 \to 5\to 1]$. MLP performs the worst, KAN achieves the best results but is slow, while GR-KAN trains faster and performs slightly worse than KAN in this experiment.
>
> |Model|L2 Error|Train Time|
> |--|--|--|
> |$w_0=10$|||
> |MLP(GELU)|2.0216e-04 |~1min|
> |GR-KAN| 6.3909e-06 |~4min|
> |KAN|1.6125e-08 |~20min|
> |$w_0=50$|||
> |MLP(GELU)|1.1805e-01 |~1min|
> |GR-KAN|5.2515e-02 |~4min|
> |KAN| 3.7762e-02|20min|
>
>
>
> The results has been incorporated in the revised paper `Appendix B`.

---

> > ### Comment · Reviewer_DyMH · 2024-11-25
> >
> > I thank the authors for the detailed response and I appreciate their efforts. I think that the contribution is now more sound and the additional details have improved the quality of the work. I still have some curiosity about the ablation study, as some relevant combinations are missing (notably only rational function), nonetheless, also being positively polarized by other reviews, I am happy to increase my rating. I think that a viable implementation of KANs can open the way for more research into this topic.

---

> > > ### Author Response · Authors · 2024-11-25
> > >
> > > We are extremely grateful for the positive feedback!
> > >
> > > If there are any other `missing combinations` the reviewer is curious about, please let us know, and we will do our best to test them.

---

### Official Review · Reviewer_cLED · 2024-11-01

**Soundness:** 3
**Presentation:** 4
**Contribution:** 3
**Rating:** 8
**Confidence:** 4

**Summary:**

This work presents the Komlogorov-Arnold Transformer, a transformer architecture that replaces the typical transformer layer's concluding 2-layer MLP with (modified) Kolmorogov-Arnold layers. The authors explain why a simple substitution of KAN layers will fail to both perform well and to scale with the hidden dimension of the network, and provide fixes (via different base functions, weight sharing across grouped edges, and improved initialization) that improve performance and speed. Improvements are shown in experiments across a variety of machine learning tasks in vision domains and beyond.

**Strengths:**

1) The paper is generally well written: it has a clear narrative, each new section is well-motivated, and for the most part, technical discussions are nicely balanced with takeaways and intuition.
2) The technical analysis and contributions of the paper directly lead to practical improvements. The entire chain of derivations/modifications creatively connect different lines of research (KAN networks + rational neural networks, for example) and add some new techniques as well. In particular, I found the re-expression of the grouped parameters in Section 4.3 into a specialized MLP to be quite elegant.
3) The experiments are comprehensive, all of the important training details are made available, and the results show that using these layers does lead to improvement in accuracy with relatively little increase in training time.

**Weaknesses:**

1) I felt some details about the method and novelty were missing. In my opinion this is mostly clearly true in the initialization section (which seems to be where a lot of the improvements/convergence come from). A lot of Section 4.4 is spent showing that choosing these initialization values is difficult and non-obvious - which I agree with! - but then there is no space left to actually show how you pick values for a and b (I know you say that you "determine a and b such that F fits established activations", but a small algorithm or set of equations would do a lot to disambiguate this).

2) The suite of experimental results is impressive, but there are comparatively few ablations studies. I am left wondering which parts of your method are responsible for the improvement over standard ViTs. This is particularly disappointing because you clearly break down your improvements step-by-step, so it is easy to imagine what such an ablation would look like. I realize that some of your modifications are meant to address scalability, so a large set of experiments might be infeasible. But something like accuracy & total training time of a ViT-T with different parts of your method (i.e., with/without your initialization scheme, with/without grouping, etc) seems like it should be possible.

3) Similarly to the above: after switching the base functions to rationals, the GR-KAN layer in practice resembles the rational layers from Boulle et al much more than it resembles the original KAN layers. But there are no direct comparisons to those; I would also find this experiment useful.

More minor pieces of feedback:

4) On L248: I don't think it's quite right to say that this gradient computation is "Similar to Monila et al"; it is identical to their calculation. I also don't think the details of this calculation are used anywhere later in the paper, unless I am mistaken, so they could also be relegated to the appendix.
5) Table 1: This caption could use more detail - is this just the forward pass, or forward and backward passes? Also, a comparison to an analogously-sized MLP layer would be additionally useful.
6) L433: "ViT-L + KAN fails to converge" - I think this should be ViT-B?
7) Figure 3: I think I understand what this figure is trying to communicate (i.e., that the choice of values for $a$ and $b$ do a good job of fitting the activation functions), but in practice it just ends up looking like... graphs of activation functions. Something quantitative would probably be more effective is communicating how well the rationals approximate the activation functions, and it would take up less space as well.

**Questions:**

1) Can you explain very specifically the novelty of your method? You break your contribution into three pieces, can you say which portions of these are novel and which are not?
2) Much of the presentation about scalability is presented through FLOPs computation. Is there a reason that you chose this representation, rather than actual training/inference time?
3) A major part of this work is the CUDA implementation of the layer. How do you plan to make this available? Will the layer also be eventually made available through frameworks like pytorch or jax? I appreciate the work that goes into a direct CUDA implementation, but I think it is crucial that there are plans for wide availability in popular frameworks in order to have this work translate to impact in the community.

---

> ### Author Response · Authors · 2024-11-21
> **Response to Reviewer cLED (Part 1)**
>
> We feel incredibly fortunate to have R-cLED's thoughtful and valuable suggestions.
>
> `>>> Q1` **Details for Initializing $a_m,b_n$**
>
> `>>> A1` Great question. Given a ground-truth function $g(\cdot)$ and a parameterized rational function $F(\cdot;\{a_m\},\{b_n\})$, we run a linear least square to determine $\{a_m\},\{b_n\}$. Specifically, we optimize the following function:
> $$\min_{\{a_m\},\{b_n\}} \frac{1}{2} \sum_{i=1}^N (g(x_i)-F(x_i;\{a_m\},\{b_n\}))^2 $$
>
> We uniformly sample 1000 points $x_i$ from the interval $[-3,3]$. $\{a_m\},\{b_n\}$ are randomly initialized. In practice, we solve this using the Levenberg-Marquardt algorithm, available in the MINPACK package.
>
> In the end, this is easily done by calling `scipy.optimize.curve_fit`. We have added and discription in the revised manuscript `Appendix G`.
>
>
> `>>> Q2` **Ablation Study**
>
> `>>> A2` We sincerely appreciate the feedback. As suggested, we conduct a new ablation study on the three factors presented in the paper, including 1) rational base function 2) group-wise weight sharing and 3) proper initialization.
>
> We conducted experiments using the KAT-Tiny model on the ImageNet dataset, training for 300 epochs
>
> - **Ablation 1: Base function**:  We replace rational base function by B-spline used in KAN. Because no pure CUDA implementation for B-spline, we implement in `torch` with De Boor-Cox algorithm.
>
>     The choice of the base function has a minor impact on performance, but significantly affects runtime.
>     According to `Exp 2` and `Exp 5`, the rational function slightly outperforms the B-spline in terms of performance. While the difference in MAC count is negligible, the B-spline implementation in `torch` runs significantly slower than our pure CUDA implementation.
>
> - **Ablation 2: Group-wise computation**: In this experiment, group-wise weight sharing is replaced with distinct parameters for each channel.
>
>     According to `Exp 4` and `Exp 6`, group-wise computation plays a crucial role in efficiency. Sharing parameters within groups slightly reduces accuracy from 74.8 to 74.6. However, it significantly reduces training time from 38 hours to 12 hours.
>
> - **Ablation 3: Initialization**: In this experiment, the variance preserving initialization is replaced by `torch` default initialization.
>
> As shown in `Exp 6` and `Exp 5`, initialization is critical for good performance. Without proper initialization, terms of different orders were initialized at similar scales, causing instability. This issue was more important for rational functions than for B-splines, as evidenced by the performance differences between `Exp 2` and `Exp 5`.
>
>
> |Exp ID | Rational | Group | Initiation | Top-1| Train Time |
> |--|--|--|--|--|--|
> |1| &#10008; | &#10008; | &#10008; | 64.9| 43h |
> |2| &#10008; | &#10004; | &#10004; | 73.0 | 20h |
> |3| &#10004; | &#10008; | &#10004; | 74.3 | 38h |
> |4| &#10004; | &#10004; | &#10008; | 53.2| **12h** |
> |5| &#10004; | &#10004; | &#10004; | **74.6**| **12h** |
>
>
> `>>> Q3` **Comparing to (Boulle et al.)**
>
> `>>> A3` Thanks the reviewer for the question. In fact, we have compared with (Boulle et al.) in our paper, since (Boulle et al.) uses **exactly the Padé Activation Unit (PAU)**. This is shown in `Table 6` as part of the ablation study, where KAT outperforms PAU.
>
>
> `>>> Q4` **Gradient Calculation (Monila et al.)**
>
> `>>> A4` We completely agree. The derivations are indeed the same. As suggested, we have moved `Equation 10` to the `Appendix F` in our revised version.

---

> ### Author Response · Authors · 2024-11-21
> **Response to Reviewer cLED (Part 2)**
>
> `>>> Q5` **Table 1 Caption**
>
> `>>> A5` We appreciate the suggestion. `Table 1` is the **forward pass FLOPs** for each edge in KAN, using different non-linear functions. We have revise in the paper.
>
> However, comparing this to an analogously-sized MLP is not straightforward because **MLPs do not apply non-linear functions on each edge**; instead, they add a non-linear function at the end. A more meaningful comparison is the overall FLOPs, which is already provided in `Table 2`.
>
> `>>> Q6` **ViT-B**
>
> `>>> A6` We have revise the `ViT-L` to `ViT-B`. Thanks.
>
> `>>> Q7` **Fig 3**
>
> `>>> A7` Thanks for the suggestion. We have added the Mean Squared Error (MSE) to the `Fig 3` to illustrate how well the rational functions approximate the activation functions.
>
> `>>> Q8` **Key novelty**
>
> `>>> A8` Great question. We believe our main novelty lie in presenting a **framework-wise solution** for integrating KAN into transformers. Additionally, we consider the **analysis** and techniques we developed to be of considerable new.
>
> - **Framework Novelty**. We are the first to successfully integrate KAN into a transformer and make it work. The replacement is simple, but no one has make this replacement sucess.
> - **Analysis Novelty**. We are the first to provide a quantitative analysis on why KAN lacks scalability. The challenges discussed in `Section 3` are introduced for the first time within the context of KAN design.
> - **Techniqueal Novelty**: Our core technical novelty lies in the development of a **group-wise** computation for activation function. To the best of our knowledge, it has not been introduced before. Besides, the application of rational functions to KAN and the initialization under such case is relatively new, although some aspects draw inspiration from prior research
>
>
> `>>> Q9` **FLOPs vs. Runtime**
>
> `>>> A9` Thanks for the question. Both metrics work for comparison, but *FLOPs are more appropriate* here because the **implementation difference**.
>
> FLOPs measure number of operations independent of implementation. In contrast, runtime comparisons can be misleading unless the code is similarly optimized.
>
> For example, The original KAN uses inefficient NumPy code, whereas we optimized GR-KAN on CUDA. GR-KAN can be $100\times$ faster on GPU, but that’s not a fair comparison. Likewise, comparing GR-KAN’s runtime to a `torch` MLP isn’t fair, as `torch` benefits from extensive optimization by thousands of engineers.
>
> Given these factors, FLOPs offer a more meaningful and consistent basis for comparison at this stage.
>
>
> `>>> Q10` **Operational plan for the project**
>
> `>>> A10` This is the right question to ask. Currently, we provide the code as a C++ extension for PyTorch, making it usable in `torch` for various tasks.
>
> We’re also in contact with the maintainers of the `Transformers` and `timm` libraries. We are right now working on integrating KAT into their codebase.

---

> > ### Comment · Reviewer_yXCb · 2024-11-26
> >
> > I would like to thank the authors for their response! I have read through all the reviewers comments and author rebuttals and responses. I am satisfied with the responses. I would also like to appreciate the authors for extensive additional experiments and clarifications provided. I have no further questions, clarifications or requirements. I will retain the scores.

---

> > > ### Author Response · Authors · 2024-11-26
> > >
> > > Thank you so much for your kind words and for taking the time to review our responses and experiments.
> > >
> > > Your feedback and support mean a lot to us!

---

> > ### Comment · Reviewer_cLED · 2024-11-27
> >
> > Thanks for this thoughtful response. This both clarified some of my misunderstandings and provided some very helpful details. I am happy to raise my review's rating from 6 to 8.

---

### Official Review · Reviewer_h5qV · 2024-11-03

**Soundness:** 3
**Presentation:** 4
**Contribution:** 3
**Rating:** 8
**Confidence:** 2

**Summary:**

The paper introduces the Kolmogorov–Arnold Transformer (KAT), where they replaces MLP layers in transformer with Kolmogorov-Arnold Network (KAN) layers. B-spline functions in KANs were replaced with rational functions to improve compatibility with modern GPUs. Group KAN was used to reduce computations and variance preserving initialization was employed. Introduces Group-rational KANs for intergation into ViT. Extensive experiments on image classsification has shown improved performance of GR-KAN.

**Strengths:**

Indentified issues with integrating KANs with ViT.
Overcame the recursive computation requirement for using B spline curves in KANs and replaced them with faster rational functions.
Introduced variance preserving initialisation to stabilize training.
Created an efficient an fast CUDA implementation of rational function for faster running in GPU
Improved performance gains by grouping and sharing parameters
Extensive experimetation is provided

**Weaknesses:**

Replacing MLP in transformers with KAN is an obvious choice and does not warrant any novelty.
phi_in matrix in eq. 2  Top row right element should show phi_1,d instead of phi_1,n
In line 46, the paper says KANs require fewer parameters than MLPs, then goes on to show in Line 176-189 than KAN requires more paramters.
The violation of variance-preserving nature in higher order B-splines is not clear.
The basic novelty is the use of rational function instead of B-splines

**Questions:**

How do you explain the contradiction of saying KANs require less paramters and then saying KAN requires more parameters?

---

> ### Author Response · Authors · 2024-11-21
>
> It is a great honour to hear the feedback from Reviewer h5qV. We have answered the questions below and revised the paper accordingly.
>
> `>>> Q1` **Novelty**
>
> `>>> A1` Thanks for the question. The reviewer is right; simply replacing the MLP with KAN is straight-forward.
> However, we believe that our key contributions lie in offering a **framework-wise solution** for integrating KAN into transformers to make this replacement scalable and practical. Additionally, we consider the **analysis** and techniques we developed to be of considerably new.
>
> - **Framework Novelty**. We are the first to successfully integrate KAN into a transformer and scale it up. While the replacement is straightforward, no prior work has succeeded to do this.
> - **Analysis Novelty**. Our work introduces the first quantitative analysis of KAN's scalability challenges, as detailed in `Section 3`.
> - **Techniqueal Novelty**: Our core technical novelty lies in the development of a **group-wise** computation for activation function. To the best of our knowledge, it has not been introduced before. Additionally, the use of rational functions in KAN and our tailored initialization method are innovative, though partially inspired by earlier studies.
>
> `>>> Q2` **Typo**
>
> `>>> A2` Thank the R-h5qV for the proof-reading. We have edit the $\phi_{in}$ matrix in `Equation 2`.
>
> `>>> Q3` **Fewer or More parameters**
>
> `>>> A3` Apologies for the confusion. These two statements are not contradictory; rather, they apply to different contexts, depending on whether we fix the  **target function** or **network width**.
>
> - **Fit the same function (Theoretical)**: As mentioned in `Line 46`, when the target function is fixed, KANs theoretically need fewer parameters to represent it.
> - **Network width (Practical)**: In `Section 3`, we explain that when the network width is fixed, KANs require more parameters than standard MLPs.
>
> Thus, these points are not opposing but are valid in different contexts.
>
> `>>> Q4` **Variance for higher order B-splines**
>
> `>>> A4` Thanks for the in-depth question. In `Line 197`,`Higher-order splines exacerbate variance instability` means that increasing spline order will excessively smooth out the input signal. It reduces function variation, causing $Var[\phi(x)]$ to become smaller.
>
> - **High-order B-spline is smoothing**. The B-spline is defined recursively as:
> $$B_{i, p}(t) = \frac{t - t_i}{t_{i+p} - t_i} B_{i, p-1}(t) + \frac{t_{i+p+1} - t}{t_{i+p+1} - t_{i+1}} B_{i+1, p-1}(t)
> $$
> shows that each higher-order basis function $B_{i, p}(t)$ a weighted summation of a weighted average of two lower-order basis functions $B_{i, p-1}(t)$ and $B_{i+1, p-1}(t)$. as the order $p$ increases, the B-spline becomes wider and smoother.
>
>     In the extreme case, as $p\to \infty$, the smoothing effect causes the basis functions to become nearly uniform across the domain, i.e., $B_{i, p}(t)\approx C$, a constant. Consequently, as, when $p\to \infty$, the variance of the output converges to zero, $Var[\phi(x)]\to 0$. This extreme smoothing leads to instability in the activation variance, as it effectively flattens all variations.
>
> We have included this explanation in `Appendix D` for further clarification.
>
>
> `>>> Q5` **Basic Novelty**
>
> `>>> A5` We appreciate the comment. We humbly believe our novelty extends beyond simply replacing B-splines with rational functions. It lies in the **complete set of new analysis and the solutions**, all introduced for the first time in the KAN context:
>
> 1. **Rational function**. We analyze the limitations of B-splines and replace them with rational functions, including a new CUDA implementation.
> 2. **Group Computation**. We identify inefficiencies in KAN, where each edge requires distinct parameters and computations. To address this, we share parameters within edge groups, reducing computational costs.
> 3. **Initialization**. We address the instability of previous initialization methods for KAN, proposing a new approach that supports the training of larger models.
>
> These contributions are interdependent and essential. Without (1), the implementation would be slow; without (2), parameter size would be excessive; and without (3), training would not converge.
>
> `>>> Q6` **Contradiction of arguments**
>
> `>>> A6` We truly thank you for the question. Please see `A3`.

---

> > ### Comment · Reviewer_h5qV · 2024-11-26
> > **Response to authors**
> >
> > I am satisfied by the authors comments.

---

> > > ### Author Response · Authors · 2024-11-26
> > >
> > > Thank the reviewer again for your kind and encouraging comments!
> > >
> > > Best!

---

### Official Review · Reviewer_yXCb · 2024-11-04

**Soundness:** 3
**Presentation:** 3
**Contribution:** 3
**Rating:** 6
**Confidence:** 4

**Summary:**

There has been a resurgence in Kolmogorov Arnold Networks (KANs) recently as an effective alternative to MLPs. This work carefully studies the major issues with scaling of the standard KAN architecture and proposes suitable modifications to create what they define as Group-Rational KAN (GR-KAN). The GR-KAN is then used as a replacement for the MLP layer in the transformer architecture in this work, which is then called the Kolmogoorv Arnold Transformer (KAT). The utility of the KAT architecture is demonstrated through experiments on various vision tasks such as Image Recognition, Object Detection, Instance Segmentation and Semantic Segmentation.

**Strengths:**

S1) Existing works using KANs in-place of MLP modules in various models (such as feedforward NNs, CNNs, Transformers etc) have had limited success due to various issues. This work goes a step further and proposes an alternative to the original KAN architecture called Group Rational (GR-KAN) solving the issues for inefficiencies in base functions, parametrizations and base initializations reducing the parameters and FLOPs from the original KAN architecture.

S2) The GR-KAN employs rational activations as the base function which are efficient and suitable for GPU computation replacing the B-Splines which are not optimized for GPU or parallel processing. (which is one of the core strengths of this work amongst related KAN literature)

S3) The KAT architecture improves the performance over standard transformer architectures such as ViT within the same computational budget.

**Weaknesses:**

W1) This work doesn’t include results on using KANs in solving PDEs (as the original work did, albeit the architecture won’t be transformers for this task) and KATs for language processing (mentioned in future work). The experiments are majorly limited to vision tasks. (Table and Graph Classification results have also been included.)

W2) Ablations on the function which parametrizes the learnable activation are not included. Ablations are mainly with respect to alternative MLPs and fixed activation functions. Ablations could have included alternatives to rational functions, B-splines etc.

**Presentation/Typos/Corrections**

I believe that the *Experiments* section can be shortened or moved to appendix and the *Appendix F Discussion and Future Work* be moved to the main paper as it packs a lot of relevant quality content.

*Appendix D Section D.1*
This is a minor correction, the number of multiplications in exponent will be $\frac{m(m-1)}{2}$ and correspondingly $\frac{n(n-1)}{2}$ instead of $\frac{m(m+1)}{2}$ and $\frac{n(n+1)}{2}$ respectively.
As an example case for $m=1$, the numerator becomes $a_0 + a_1x$ which involves 0 multiplications in the exponent computation and a single multiplication from the multiplication for the coefficients.
The total multiplications will be in fact $\frac{m(m+1)}{2}$ which is $\frac{m(m-1)}{2}+m$
Note that with this change the values in the Table 1 and the appendix D will change accordingly.

**Questions:**

Q1) As per my understanding, Kolmogorov Arnold Representation Theorem and the setup mentioned here involves learning a multivariate function, why do *lines 146-147* in the *section 2.2* mention *learn a univariate functions on the edge*. The dimension of $f(x)$ would be
$d_{out}$ making it multivariate?

Q2) Do we have a theoretical understanding of the behaviour outlined in lines *316-317*? How are the parameters adjusted in case of B-splines or any other type of functions?

---

> ### Author Response · Authors · 2024-11-21
> **Thank Reviewer yXCb for the valuable comments**
>
> We sincerely thank Reviewer yXCb for valuable comments and suggestions. We have carefully incorporated them to improve our paper.
>
> `>>> Q1` **Results on more tasks**
>
> `>>> A1` We truly appreciate the suggestions. As suggested, we add experiment on solving PDE and NLP  task.
>
> - **GR-KAN for PDE**: For PDE solving, we resort to a one-dimensional damped harmonic oscillator. It is governed by the differential equation that
>
>     $$m \frac{{d^2 u}}{{dt^2}} + \mu \frac{{du}}{{dt}} + k u = 0$$
>     where $m$ is the mass. $\mu$ is the damping coefficient. $k$ is the stiffness constant. With the initial condition $u(0) = 1,u'(0) = 0,m = 1$. The exact solution is $u(t) = e^{-d \cdot t} \cdot (A \cos(\omega t + \phi))$, where $d = \mu / 2 w_0 = \sqrt{k}, w = \sqrt{w_0^2 - d^2}, \phi = \arctan(-d / w)$. We solved this problem using MLP, KAN, and GR-KAN with a network architecture of $[2 \to 5\to 1]$. KAN achieved the best accuracy but trained slowly, MLP performed the worst, and GR-KAN trained faster than KAN but performed slightly worse in this experiment.
>
>     |Model|L2 Error|Train Time|
>     |--|--|--|
>     |$w_0=10$|||
>     |MLP(GELU)|2.0216e-04 |~1min|
>     |GR-KAN| 6.3909e-06 |~4min|
>     |KAN|1.6125e-08 |~20min|
>     |$w_0=50$|||
>     |MLP(GELU)|1.1805e-01 |~1min|
>     |GR-KAN|5.2515e-02 |~4min|
>     |KAN| 3.7762e-02|20min|
>
>
>     The results have been incorporated in the revised `Appendix B`.
>
> - **KAT for NLP**: Due to the tight schedule during rebuttal, we only do some preliminary experiments. We fine-tuned ALBERT-base on two datasets from GLUE: the Stanford Sentiment Treebank (SST-2) and Multi-Genre NLI (MNLI). We replaced the MLP with the GR-KAN designed in the paper. We inherited all weights from pretrained model as described in `Section 4.4`.
>
>   The accuracy is reported below. We achieved better performance. We will add full GLUE results once they are completed.
>
>     |Model|SST-2|MNLI|
>     |--|--|--|
>     |ALBERT-Base|90.3|81.6|
>     |ALBERT-Base+KAT|**91.6**|**83.2**|
>
>
> `>>> Q2` **Ablation Study on the base function**
>
> `>>> A2` Thank the reviewer for the question. In fact, we have conducted an ablation study. ViT + KAN means that we use B-splines same as the KAN paper and compare with our KAT, as shown in `Figure 5`. However, this experiment ablated all components together.
>
> Based on the suggestion, we ran a more focused study. We re-implemented the method using radiance base functions (RBF) and B-splines, but keeping group-wise computation and initialization consistent.
>
> Note that this PyTorch implementation is still slower than our optimized CUDA version. Our Rational function delivers the best performance.
>
> | Base function | Top-1|
> |--|--|
> | RBF | 73.2 |
> | B-spline | 73.0|
> | Rational (Ours) | **74.6**|
>
>
>
> `>>> Q3` **Computation Correction**
>
> `>>> A3`We sincerely thank Reviewer yXCb for the thorough proofreading. As suggested, we have revised both  `Table 1` and the `Appendix I` as suggested.
>
> `>>> Q4` **Multivariate and Univariate**
>
> `>>> A4` Sorry for the confusion might caused. We mean "*learn a univariate function on **each** edge*". Altogether, summation of all univariate functions forms a multivariate function. This terminology is consistent with the KAN paper. We have revised the text in `Line 146` to make it clearer.
>
> `>>> Q5` **Reason for Shared Coefficient**
>
> `>>> A5` This decision is based on empirical observations. Using different coefficients for different groups, especially for denominators, slightly reduces performance. We attribute this to the sensitivity of division operations.
>
> **Hypothesis: Denominator sensitivity**. In rational functions, small changes in the denominator can cause significant output variations. Additionally, varying embedding scales across groups can introduce inconsistencies when separate coefficients are used, negatively affecting performance.
>
> We have added a formal discussion in `Appendix E` to clarify this.

---

### Author Response · Authors · 2024-11-21
**Thank all reviewers for their constructive feedback**

We sincerely thank all reviewers for their constructive feedback. We deeply appreciate the following positive comments:

- The paper provides deep technical analysis: `Reviewer yXCb, Reviewer cLED`
- The approach is scalable, fast, and general, with a fast CUDA implementation: `Reviewer yXCb, Reviewer h5qV, Reviewer DyMH`.
- The experiments are extensive: `Reviewer h5qV, Reviewer cLED, Reviewer DyMH`
- The method achieves good performance: `Reviewer yXCb`
- The writing is clear and well-supported:`Reviewer cLED, Reviewer s19p`

We will address the specific questions and concerns raised by the reviewers in the subsequent sections of this rebuttal.

---

### Meta-Review · Area_Chair_923Y · 2024-12-17

**Metareview:**

The paper introduces the Kolmogorov–Arnold Transformer (KAT), which replaces traditional MLP layers in transformers with Kolmogorov-Arnold Network (KAN) layers to enhance model performance. The authors address three main challenges: the inefficiency of B-spline functions, the computational load of unique functions for each input-output pair, and the difficulty of weight initialization.
The KAT architecture demonstrates improved performance across various vision tasks such as image recognition, object detection, and semantic segmentation. Concretely, the authors demonstrate the benefits over the base transformers ViT and DeiT in the seminal ImageNet-1k, prototyping their architecture there. The KAN achieves an improved performance with the same or less FLOPs. One major drawback of the paper is that it does not compare with other architectures, e.g., MLP-Mixers or their variants, Mamba variants, etc. There is a wealth of architectures the last few years, including papers published in all major ML conferences in 2024, which I hope the authors can include in the camera-ready paper.

**Additional Comments On Reviewer Discussion:**

During the rebuttal, the authors raised various questions, including additional experiments and applications beyond vision, which the authors have conducted. Concretely, the authors have posed questions regarding the novelty of the method, or for the limited scope of experiments. In response, the authors have addressed the questions over the novelty and included new ablation studies, or demonstrated how the KAN model can perform well in NLP or even PDEs. Again, the experiments could be further enriched by comparing with other families of models proposed for tackling PDEs recently.

---

### Decision · Program_Chairs · 2025-01-22

Accept (Poster)